# Simulation of the 1d XY model on a quantum computer

Marc Farreras[1,2], Alba Cervera-Lierta[2]

1. Departament de Física Quàntica i Astrofísica, Universitat de Barcelona, 08028 Barcelona, Spain

2. Barcelona Supercomputing Center, Plaça Eusebi Güell, 1-3, 08034 Barcelona, Spain

## Abstract

The field of quantum computing has grown fast in recent years, both in theoretical advancements and the practical construction of quantum computers. These computers were initially proposed, among other reasons, to efficiently simulate and comprehend the complexities of quantum physics. In this paper, we present the comprehensive scheme for the exact simulation of the 1-D XY model on a quantum computer. We successfully diagonalize the proposed Hamiltonian, enabling access to the complete energy spectrum. Furthermore, we propose a novel approach to design a quantum circuit to perform exact time evolution. Among all the possibilities this opens, we compute the ground and excited state energies for the symmetric XY model with spin chains of $n = 4$ and $n = 8$ spins. Further, we calculate the expected value of transverse magnetization for the ground state in the transverse Ising model. Both studies allow the observation of a quantum phase transition from an antiferromagnetic to a paramagnetic state. Additionally, we have simulated the time evolution of the state all spins up in the transverse Ising model. The scalability and high performance of our algorithm make it an ideal candidate for benchmarking purposes, while also laying the foundation for simulating other integrable models on quantum computers.

## 1 Introduction

In the first decade of the XXI century, we witnessed an explosion of the quantum computing field driven by the incredible potential that quantum computing exhibits to solve some intractable classical problems [1]. Among these challenges, one of the enduring objectives of quantum computing is the simulation of quantum systems. Although several classical strategies exist for simulating such systems [2, 3], they often prove to be inefficient when dealing with complex quantum systems. Consequently, the simulation of quantum systems demands alternative methods for efficient execution. Here, quantum computers emerge as a promising solution, since due to their quantum nature the simulation of strongly correlated systems

is the natural arena where quantum computers are expected to show a clear advantage over classical ones, as Feynman stated in Ref.[4].

Despite having undergone considerable development during the last decade, quantum computing is still in an early stage. The current state of quantum computing is known as the Noisy Intermediate-Scale Quantum (NISQ) era [5]. The NISQ era has been characterized by constrained-size quantum processors (containing 100 qubits approximately) with imperfect control over them; they are sensitive to their environment and prone to quantum decoherence and other sources of errors. In this context, it has become essential to have available methods that allow companies and researchers to test the efficiency of their new devices or to compare them. However, due to the limited number of qubits and the high error rate, to find proper algorithms for NISQ devices has become a difficult task. These algorithms need to be easy to escalate with the number of qubits and present a feasible circuit depth to avoid the large accumulation of noise.

This paper presents a circuit specifically designed for the NISQ era, offering the capability to explore intriguing phenomena such as quantum phase transitions. Our work consists of implementing a quantum circuit that performs the exact simulation of a 1-D spin chain with an XY -type interaction. We programmed a set of Python libraries that allows the implementation of the circuit for systems with a power of 2 number of qubits using Qibo [6], an open-source framework for quantum computing. Moreover, Qibo is the native language of the Barcelona Supercomputing Center quantum computer, which will allow the users to directly test this algorithm with real machines. The foundation of our work is based on Ref.[7, 8], where the steps followed to design the quantum circuit rest upon tracing and implementing the well-known transformations that solve the model analytically [9]. As a result, this technique can access the whole spectrum, enabling us to simulate any excited or thermal state and its dynamical evolution. In addition, this framework can be easily extended to other integrable models, including the Kitaev-honeycomb model [10], or to systems whose effective low-energy behavior can be suitably described by quasi-particles.

This paper is organized as follows: In Sec.2 we describe the characteristics of the XY model and solve it analytically. Moving to Sec.3, we revisit the method introduced in Ref.[7] to construct an efficient circuit that diagonalizes the XY Hamiltonian. We then present the circuit employed for simulating spin chains of $n = 4$ and $n = 8$ qubits. Next, in Sec.4 we design a quantum circuit tailored for exact time evolution. Our simulations, utilizing the proposed quantum circuit, are detailed in Sec.5. Finally, the conclusions are exposed in Sec.6 and the code is available in Ref.[11].

## 2    The $1 - D$ XY model

The XY model is derived from the Heisenberg model [12] by introducing an easy-plane anisotropy. Those models are widely used to study critical points and phase transitions of magnetic systems within the condensed matter field. The $1 - D$ XY Hamiltonian can be written as

$$\mathcal{H}_{XY} = J \left( \sum_{i=1}^{n} \frac{1+\gamma}{2} \sigma_i^x \sigma_{i+1}^x + \frac{1-\gamma}{2} \sigma_i^y \sigma_{i+1}^y \right) + \lambda \sum_{i=1}^{n} \sigma_i^z \tag{1}$$

where $n$ is the number of spins in the 1-D spin chain, $\sigma_j^i$ with $i = x, y, z$ are the Pauli matrix acting on the site $j$, $J$ determine the behavior of the ordered phase, ferromagnetic for $J < 0$ and antiferromagnetic $J > 0$, $\gamma$ is the anisotropic parameter and $\lambda$ represents the strength of the transverse magnetic field.

One important feature for which the XY model stands out is that it exhibits a quantum phase transition [13, 14]. These transitions occur at absolute zero and stem from the competition of the different terms within the Hamiltonian, regulated by a non-thermal physical parameter of the system. At zero temperature, each term presents a specific ground state, and the properties of these ground states dictate the phase of the system.

Specifically for the $1 - D$ XY model, the Hamiltonian presents three terms with ground states that exhibit different phases. The first two terms parametrized by $J$ and $\gamma$ are $\sigma_i^x \sigma_{i+1}^x$ and $\sigma_i^y \sigma_{i+1}^y$. Both by themselves correspond to the well-known Ising model, in which the ground state is ferromagnetic or antiferromagnetic, depending on the sign of $J$, and points respectively to the $x$ or $y$ axis. Contrarily, the ground state of the third term $\sigma_i^z$, parametrized by $\lambda$ is paramagnetic and points to the $z$ axis. As a result, the ground state will show ferromagnetic or antiferromagnetic behavior when $|J| > \lambda$ and the direction of the spin will be mediated by $\gamma$. However, the ground state will show paramagnetic behavior for $|J| < \lambda$. In Fig.1 there is shown the phase diagram at $T = 0$ of the 1-D XY model for $J = -1$.

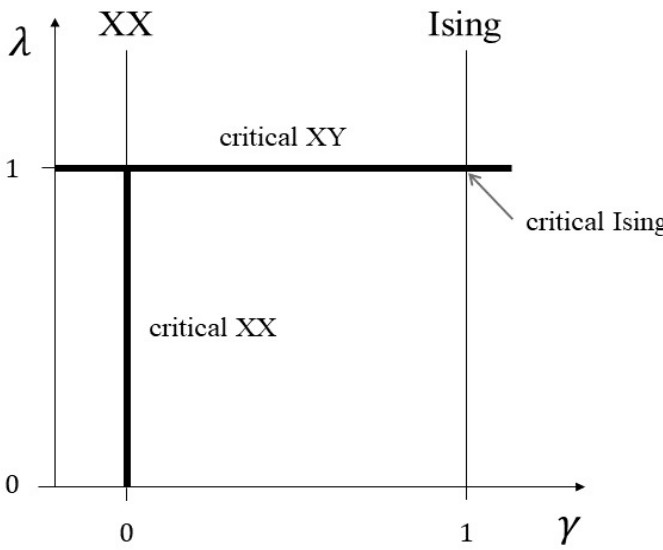

Figure 1: Phase diagram of the quantum XY model.

In the next subsections, we derive the analytical solution of the XY model. However, before starting is convenient to rewrite Eq.(1) it in terms of spin leather operators $\sigma^{+(-)}$ which increase(decrease) the projection of the third component of the spin $S_z$ by 1. The $\sigma^x$ and $\sigma^y$ operators then can be written as

$$
\begin{aligned}
\sigma^x &= \sigma^+ + \sigma^-, \\
\sigma^y &= -i\left(\sigma^+ + \sigma^-\right), \\
\sigma^z &= 2\sigma^+\sigma^- - 1.
\end{aligned}
\tag{2}
$$

Hence, the Hamiltonian from Eq.(1) becomes

$$
\mathcal{H}'_{XY} = J\left(\sum_{i=1}^{n-1}\sigma_i^+\sigma_{i+1}^- + \sigma_i^-\sigma_{i+1}^+ + \gamma\left(\sigma_i^+\sigma_{i+1}^+ + \sigma_i^-\sigma_{i+1}^-+\right)\right) + \lambda\sum_{i=1}^{n}\left(2\sigma^+\sigma^- - 1\right).
\tag{3}
$$

Furthermore, it is worth remembering some properties from the Spin $\frac{1}{2}$, which will be used later on in the next steps.

$$
\begin{aligned}
\sigma^+\left(-\sigma^z\right) &= \sigma^+, & \sigma^-\left(-\sigma^z\right) &= -\sigma^-, \\
-\sigma^z\sigma^+ &= -\sigma^+, & -\sigma^z\sigma^- &= \sigma^-.
\end{aligned}
\tag{4}
$$

## 2.1 Jordan-Wigner transformation

Generally, quantum spin objects are notoriously difficult to deal with in many-body physics because they neither fulfill fermionic nor bosonic algebra. For this reason, the first step to diagonalize XY Hamiltonian consists of applying the Jordan-Wigner transformation [15] which maps the spin operators $\sigma$ into spinless fermionic modes $c$.

The Jordan-Wigner transformation takes advantage of the similarities between fermions and spin operators. The existence of the stated similarity can be noticed by how both operators act on their respective basis, where fermionic basis $|1\rangle$ and $|0\rangle$ respectively corresponds to having one or no fermion in the state (no fermion state is also called void), while $|+\rangle$ and $|-\rangle$ means having a spin pointing up or down in the $z$ axis. As shown in Table 1, there is a clear equivalence between $|0\rangle$ and $|-\rangle$, and the same with $|1\rangle$ and $|+\rangle$.

| Fermions | Spin $\frac{1}{2}$ |
|---|---|
| $c^\dagger \lvert 0 \rangle = \lvert 1 \rangle$ | $\sigma^+ \lvert - \rangle = \lvert + \rangle$ |
| $c^\dagger \lvert 1 \rangle = 0$ | $\sigma^+ \lvert + \rangle = 0$ |
| $c \lvert 0 \rangle = 0$ | $\sigma^- \lvert - \rangle = 0$ |
| $c \lvert 1 \rangle = \lvert 0 \rangle$ | $\sigma^- \lvert + \rangle = \lvert - \rangle$ |

Table 1: Fermionic and Spin operator's behavior when acting in their respective basis.

However, there is also an important difference between them, their commutation relationships. The commutation relationship followed by $\sigma^{+(-)}$ and $\sigma^-$ operators are

$$
\begin{aligned}
[\sigma_j^+, \sigma_i^-] &= 0 \quad i \neq j, \\
\{\sigma_i^+, \sigma_j^-\} &= I \quad i = j,
\end{aligned}
\tag{5}
$$

while the operators $c$ and $c^\dagger$ obey the fermionic algebra $\{c_i^\dagger, c_j\} = \delta_{ij}$.

The brilliant idea that Jordan and Wigner came up with to solve this problem was to add an operator, called string operator $e^{\pi i \sum_{j=1}^{i-1} \sigma_j^+ \sigma_j^-}$, which takes into count how many $\lvert + \rangle$ states or fermionic particles are present in our system and helps to map the spin operators to fermionic operators.

Therefore, Jordan-Wigner transformation is defined as

$$
\begin{aligned}
c_i^\dagger &= \sigma_i^+ e^{-\pi i \sum_{j=1}^{i-1} \sigma_j^+ \sigma_j^-}, \\
c_i &= e^{\pi i \sum_{j=1}^{i-1} \sigma_j^+ \sigma_j^-} \sigma_j^-, \\
c_i^\dagger c_i &= \sigma_j^+ \sigma_j^-.
\end{aligned}
\tag{6}
$$

Where the $c$ and $c_i^\dagger$ are the new spinless fermionic operators. Note that $\sigma_j^+ \sigma_j^-$ and $\sigma_i^+ \sigma_i^-$ commute

$$
\begin{aligned}
[\sigma_i^+ \sigma_i^-, \sigma_j^-] &= -\delta_{ij} \sigma_j^-, \\
[\sigma_i^+ \sigma_i^-, \sigma_j^+] &= \delta_{ij} \sigma_j^+, \\
[\sigma_i^+ \sigma_i^-, \sigma_j^+ \sigma_j^-] &= \sigma_j^+ [\sigma_i^+ \sigma_i^-, \sigma_j^-] + [\sigma_i^+ \sigma_i^-, \sigma_j^+] \sigma_j^- = 0.
\end{aligned}
\tag{7}
$$

Therefore,

$$
e^{\pm i\pi \sum_{j=n}^m \sigma_j^+ \sigma_j^-} = \prod_{j=n}^m e^{\pm i\pi \sigma_j^+ \sigma_j^-}.
\tag{8}
$$

The next step is to develop the exponential operator

$$
\begin{aligned}
e^{\pm i\pi \sigma_j^+ \sigma_j^-} &= \sum_{l=0}^\infty \frac{1}{l!} (\pm i\pi)^l \left(\sigma_j^+ \sigma_j^-\right)^l = 1 + \sum_{l=1}^\infty \frac{1}{l!} (\pm i\pi)^l \sigma_j^+ \sigma_j^- = \\
&= 1 + \left(e^{\pm i\pi} - 1\right) \sigma_j^+ \sigma_j^- = 1 - 2\sigma_j^+ \sigma_j^- = -\sigma_j^z.
\end{aligned}
\tag{9}
$$

In consequence, sometimes the Wigner-Jordan transformation is also written as

$$
\begin{aligned}
c_i^\dagger &= \sigma_i^+ \left(\prod_{l=1}^{i-1} -\sigma_l^z\right), \\
c_i &= \left(\prod_{l=1}^{i-1} -\sigma_l^z\right) \sigma_j^-, \\
c_i^\dagger c_i &= \sigma_j^+ \sigma_j^-.
\end{aligned}
\tag{10}
$$

For the sake of the study's further development, it is interesting to write down the inverse transformation

$$\sigma_i^+ = c_i^\dagger \left( \prod_{l=1}^{i-1} -\sigma_l^z \right) = c_i^\dagger e^{\pi i \sum_{j=1}^{i-1} \sigma_j^+ \sigma_j^-},$$

$$\sigma_i^- = \left( \prod_{l=1}^{i-1} -\sigma_l^z \right) c_i = e^{-\pi i \sum_{j=1}^{i-1} \sigma_j^+ \sigma_j^-} c_i, \tag{11}$$

$$\sigma_i^z = 2c_i^\dagger c_i - 1.$$

Now the transformed spin operators obey the canonical fermion algebra

$$\{c_i, c_j^\dagger\} = \delta_{ij}, \ \ \{c_i, c_j\} = 0, \ \ \{c_i^\dagger, c_j^\dagger\} = 0, \tag{12}$$

as it is shown in Ref.[16].

Subsequently, let's derive some useful relations. Using Eq.(12) we can to compute the following commutator

$$[c_i^\dagger c_i, c_j^\dagger c_j] = c_i^\dagger c_i c_j^\dagger c_j - c_j^\dagger c_j c_i^\dagger c_i = c_i^\dagger c_i c_j^\dagger c_j - (-1)^4 c_i^\dagger c_i c_j^\dagger c_j = 0 \ if \ i \neq j. \tag{13}$$

For the case $i = j$ the commutator is also 0. Additional useful commutator relations are

$$[c_i^\dagger c_i, c_j] = c_i^\dagger c_i c_j - c_j c_i^\dagger c_i = c_i^\dagger c_i c_j - \delta_{ji} c_i +$$
$$+ c_i^\dagger c_j c_i = c_i^\dagger c_i c_j - \delta_{ji} c_i - c_i^\dagger c_i c_j = -\delta_{ji} c_i,$$
$$[c_i^\dagger c_i, c_j^\dagger] = \delta_{ji} c_i^\dagger, \tag{14}$$
$$(c_i^\dagger c_i)^2 = c_i^\dagger c_i.$$

Applying the different properties derived from the expressions before and $c_i c_i = 0$, one can compute

$$\{1 - 2c_i^\dagger c_i, c_i\} = c_i - 2c_i^\dagger c_i c_i + c_i - c_i 2c_i^\dagger c_i =$$
$$= 2c_i - 2c_i c_i^\dagger c_i = 2c_i - 2(c_i^\dagger c_i c_i + c_i) = 0, \tag{15}$$
$$\{1 - 2c_i^\dagger c_i, c_i^\dagger\} = 0.$$

Now, using the properties from Eq.(14) and Eq.(15), one can compute the commutation relationship between the string operator described in Eq.(8) and the new fermionic operators

$$[e^{\pm i\pi \sum_{j=n}^m c_j^\dagger c_j}, c_i] = [e^{\pm i\pi \sum_{j=n}^m c_j^\dagger c_j}, c_i^\dagger] = 0, \ \ i \notin [n, m],$$
$$\{e^{\pm i\pi \sum_{j=n}^m c_j^\dagger c_j}, c_i\} = \{e^{\pm i\pi \sum_{j=n}^m c_j^\dagger c_j}, c_i^\dagger\} = 0, \ \ i \in [n, m]. \tag{16}$$

**Jordan-Wigner transformation in the XY model**

Here, we apply the Jordan-Wigner transformation into the elements that appear in the XY Hamiltonian, from Eq.(3). For the $\sigma_i^+ \sigma_{i+1}^+$ term,

$$\sigma_i^+ \sigma_{i+1}^+ = c_i^\dagger e^{\pi i \sum_{j=1}^{i-1} \sigma_j^+ \sigma_j^-} c_{i+1}^\dagger e^{\pi i \sum_{j=1}^{i} \sigma_j^+ \sigma_j^-} = c_i^\dagger e^{\pi i \sum_{j=1}^{i-1} \sigma_j^+ \sigma_j^-} e^{\pi i \sum_{j=1}^{i} \sigma_j^+ \sigma_j^-} c_{i+1}^\dagger =$$
$$= c_i^\dagger (-\sigma_i^z) c_{i+1}^\dagger = c_i^\dagger \left( -2c_i^\dagger c_i + 1 \right) c_{i+1}^\dagger = c_i^\dagger c_{i+1}^\dagger, \tag{17}$$

where in the last step we have taken in count that $c_i^\dagger c_i^\dagger = 0$.

Likewise, one can calculate the $\sigma_i^- \sigma_{i+1}^-$ term

$$\sigma_i^- \sigma_{i+1}^- = e^{-\pi i \sum_{j=1}^{i-1} \sigma_j^+ \sigma_j^-} c_i e^{-\pi i \sum_{j=1}^{i} \sigma_j^+ \sigma_j^-} c_{i+1} = c_i \left( -2c_i^\dagger c_i + 1 \right) c_{i+1} =$$
$$= c_i \left( -2 \left( -c_i c_i^\dagger + 1 \right) + 1 \right) c_{i+1} = -c_i c_{i+1} = c_{i+1} c_i. \tag{18}$$

Lastly, the $\sigma_i^+ \sigma_{i+1}^-$ and $\sigma_i^- \sigma_{i+1}^+$ can be transformed following the same procedure described above

$$\sigma_i^+ \sigma_{i+1}^- = c_i^\dagger c_{i+1},$$
$$\sigma_i^- \sigma_{i+1}^+ = c_{i+1}^\dagger c_i, \tag{19}$$

**Boundary conditions**

Until now, we have not mentioned anything about what happens in the boundary terms $\sigma_{n+1}$. Given the finite nature of our simulations, it becomes imperative to establish certain boundary conditions for our system. Specifically, we've implemented periodic boundary conditions (PBC). However, it's worth noting that we've opted for a direct application of PBC within the fermionic space. This choice translates to the relationship between fermionic operators, namely, $c_n c_{n+1} = c_n c_1$.

To add this term to our XY Hamiltonian, first, it has to be mapped into the spin space using the Jordan-Wigner transformation. Unfortunately, this transformation maps the PBC to PBC or antiperiodic boundary condition (APBC) depending on whether the system has an odd or even number of particles or $|+\rangle$ states. Consequently, the boundary term of our Hamiltonian must present this parity dependence to correctly be mapped into PBC in the fermionic space, this can be achieved using the $\sigma_1^y \sigma_2^z \cdots \sigma_{n-1}^z \sigma_n^y$ and $\sigma_1^x \sigma_2^z \cdots \sigma_{n-1}^z \sigma_n^x$ terms from Eq.(1). Then the Hamiltonian simulated in this work reads

$$\mathcal{H}_{XY} = J \left( \sum_{i=1}^{n-1} \frac{1+\gamma}{2} \sigma_i^x \sigma_{i+1}^x + \frac{1-\gamma}{2} \sigma_i^y \sigma_{i+1}^y \right) + \lambda \sum_{i=1}^{n} \sigma_i^z \tag{20}$$

$$+ J \frac{1+\gamma}{2} \sigma_1^y \sigma_2^z \cdots \sigma_{n-1}^z \sigma_n^y + J \frac{1-\gamma}{2} \sigma_1^x \sigma_2^z \cdots \sigma_{n-1}^z \sigma_n^x. \tag{21}$$

The first two terms correspond to the $1 - D$ XY Hamiltonian, whereas the last two terms belong to the boundary conditions. These boundary terms can be substituted with the conventional periodic terms $\sigma_n^x \sigma_1^x$ and $\sigma_n^y \sigma_1^y$ for states with an even number of spins pointing up, and the same terms with a negative sign for states with an odd number of spins pointing up. It is worth keeping in mind that even the Hamiltonian we are working on is not strictly the same as the XY model, in the thermodynamic limit the boundary conditions do not play any role and we recover the same results.

Now we will demonstrate that when the Jordan Wigner transformation is applied to this term, the PBC is recovered for the fermionic operators. First, we need to write the $\sigma_1^y \sigma_2^z \cdots \sigma_{n-1}^z \sigma_n^y$ and $\sigma_1^x \sigma_2^z \cdots \sigma_{n-1}^z \sigma_n^x$ using the $\sigma^+$ and $\sigma^-$ operators

$$\begin{aligned}
\sigma_1^y \sigma_2^z \cdots \sigma_{n-1}^z \sigma_n^y &= -\sigma_1^+ \sigma_2^z \cdots \sigma_{n-1}^z \sigma_n^+ + \sigma_1^+ \sigma_2^z \cdots \sigma_{n-1}^z \sigma_n^- + \\
&\quad + \sigma_1^- \sigma_2^z \cdots \sigma_{n-1}^z \sigma_n^+ - \sigma_1^- \sigma_2^z \cdots \sigma_{n-1}^z \sigma_n^-, \\
\sigma_1^x \sigma_2^z \cdots \sigma_{n-1}^z \sigma_n^x &= \sigma_1^+ \sigma_2^z \cdots \sigma_{n-1}^z \sigma_n^+ + \sigma_1^+ \sigma_2^z \cdots \sigma_{n-1}^z \sigma_n^- + \\
&\quad + \sigma_1^- \sigma_2^z \cdots \sigma_{n-1}^z \sigma_n^+ + \sigma_1^- \sigma_2^z \cdots \sigma_{n-1}^z \sigma_n^-.
\end{aligned} \tag{22}$$

Next, the Jordan-Wigner transformation is applied to the different terms that appear in the above expression using the properties shown in Eq.(4), $\sigma^z \sigma^z = 1$ and we will restrict our system to have an even number of qubits ($n$). First let's compute the term $\sigma_1^+ \sigma_2^z \cdots \sigma_{n-1}^z \sigma_n^+$,

$$\sigma_1^+ \sigma_2^z \cdots \sigma_{n-1}^z \sigma_n^+ = c_1^\dagger \sigma_2^z \cdots \sigma_{n-1}^z \left( \prod_{l=1}^{n-1} -\sigma_l^z \right) c_n^\dagger = \tag{23}$$
$$= c_1^\dagger \left( -\sigma_1^z \right) (-1)^{n-2} c_n^\dagger = c_1^\dagger \left( -2 c_1^\dagger c_1 + 1 \right) c_n^\dagger = c_1^\dagger c_n^\dagger.$$

The rest of the terms can be computed following the same steps, the results are summarized in the following expressions

$$\begin{aligned}
\sigma_1^+ \sigma_2^z \cdots \sigma_{n-1}^z \sigma_n^+ &= c_1^\dagger c_n^\dagger, \\
\sigma_1^+ \sigma_2^z \cdots \sigma_{n-1}^z \sigma_n^- &= c_1^\dagger c_n, \\
\sigma_1^- \sigma_2^z \cdots \sigma_{n-1}^z \sigma_n^+ &= c_n^\dagger c_1, \\
\sigma_1^- \sigma_2^z \cdots \sigma_{n-1}^z \sigma_n^- &= c_n c_1,
\end{aligned} \tag{24}$$

Subsequently, the boundary term reads

$$\mathcal{H}_{BC} = \frac{1+\gamma}{2} \sigma_1^y \sigma_2^z \cdots \sigma_{n-1}^z \sigma_n^y + \frac{1-\gamma}{2} \sigma_1^x \sigma_2^z \cdots \sigma_{n-1}^z \sigma_n^x =$$
$$= c_1^\dagger c_n + c_n^\dagger c_1 + \gamma \left( c_n^\dagger c_1^\dagger + c_1 c_n \right) = c_{n+1}^\dagger c_n + c_n^\dagger c_{n+1} + \gamma \left( c_n^\dagger c_{n+1}^\dagger + c_{n+1} c_n \right),$$

where now is easy to see that this Hamiltonian fulfills the PBC in the fermionic space.

## 2.2 Fermionic Fourier Transform (fFT)

The result of assembling all the solutions detailed in the previous sections gives the Hamiltonian corresponding to the XY model but now is quadratic in fermionic annihilation and creation operators $c$ and $c^\dagger$ instead of quadratic in spin operator $\sigma^+$ and $\sigma^-$

$$\mathcal{H}_{JW} = J \sum_{i=1}^{n} \left( c_i^\dagger c_{i+1} + c_{i+1}^\dagger c_i + \gamma \left( c_i^\dagger c_{i+1}^\dagger + c_{i+1} c_i \right) \right) + \lambda \sum_{i=1}^{n} \left( 2 c_i^\dagger c_i - 1 \right). \tag{25}$$

The quadratic Hamiltonian in fermionic annihilation and creation operators appears in more condensed matter systems notably exemplified in the Hubbard model [17]. Diagonalizing this type of Hamiltonian is a well-established procedure, leading us to the subsequent phase: the fermionic Fourier transform (fFT).

In the second quantization, the Fourier transform is defined as:

$$c_j = \frac{1}{\sqrt{N}} \sum_{k=-\frac{n}{2}+1}^{\frac{n}{2}} b_k e^{i\frac{2\pi k}{n} j}, \quad c_j^\dagger = \frac{1}{\sqrt{N}} \sum_{k=-\frac{n}{2}+1}^{\frac{n}{2}} b_k^\dagger e^{-i\frac{2\pi k}{n} j}, \tag{26}$$

where $b_k^\dagger$ and $b_k$ are the creation and annihilation operators of the fermionic Fourier modes.

The discrete $k$ values are acquired establishing the translational invariance of the system by PBC

$$|x + n\rangle = |x\rangle,$$
$$\sum_k e^{i\frac{2\pi k}{n}(x+n)} |k\rangle = \sum_{k'} e^{i\frac{2\pi k'}{n}(x)} |k'\rangle. \tag{27}$$

Then, we multiply at both sides by $\langle k|$, and applying $\langle k|k'\rangle = \delta_{k,k'}$

$$e^{i\frac{2\pi k}{n}(x+n)} = e^{i\frac{2\pi k}{n}(x)} \rightarrow e^{i\frac{2\pi k n}{n}} = 1,$$
$$\frac{2\pi k n}{n} = 2\pi m \rightarrow k = m, \tag{28}$$

where $m$ is an integer. Because the number of qubits ($n$) is even, as mentioned in Section 2.1, we can choose our $k$ values to be

$$k = -\frac{n}{2} + 1, -\frac{n}{2} + 2, ..., -1, 0, 1, ..., \frac{n}{2} - 1, \frac{n}{2}. \tag{29}$$

In the case where the number of qubits is odd, from Eq.(23) it can be seen that an extra "$-$" sign appears in the final result obtaining APBC $-c_1^\dagger c_n^\dagger$. Then applying translational invariance we get that the $k$ possible values are the same as in the previous case. As surprising as it may seem, one can expect this result if one thinks in terms of sinusoidal functions. If the period of the sinusoidal function is $L$ we will recover in $x = 0$ the same result as in $x = L$, hence we have PBC. Nonetheless, if our lattice ends in $x = \frac{L}{2}$ then we will have the same absolute value in $x = 0$ and $x = \frac{L}{2}$ but with a different sign. As a result, we have APBC. In the end, the $n$ odd case for APBC must have the same $k$ values as $2n$ with PBC.

Notice that even though we will be only focusing on the even number of qubits case, the procedure followed for the odd case will be equivalent to the one we will describe for the even case. One can find more information about the general case and boundary conditions in Ref.[18].

### Fermionic Fourier Transform in the XY model

Before starting to compute the new terms of the XY Hamiltonian, it is worth remembering the following properties of the FT

$$\frac{1}{N} \sum_k e^{i\frac{2\pi k}{n}(j-j')} = \delta_{j,j'}, \quad \frac{1}{N} \sum_j e^{i\frac{2\pi (k-q)}{n} j} = \delta_{k,q}, \tag{30}$$

where $\delta_{j,j'}$ and $\delta_{k,q}$ are Kronecker deltas, which are 0 when $j \neq j'$ or $k \neq q$ and 1 when are equals.

In order to calculate the FT of our Hamiltonian, the different terms that appear in Eq.(25) will be computed independently.

$$\sum_{j=1}^{n} c_j^\dagger c_{j+1} = \sum_{j=1}^{n} \left( \frac{1}{\sqrt{n}} \sum_{k=-\frac{n}{2}+1}^{\frac{n}{2}} b_k^\dagger e^{-i\frac{2\pi k}{n}j} \right) \left( \frac{1}{\sqrt{n}} \sum_{k'=-\frac{n}{2}+1}^{\frac{n}{2}} b_{k'} e^{i\frac{2\pi k'}{n}(j+1)} \right) =$$

$$= \sum_k \sum_{k'} b_k^\dagger b_{k'} \frac{1}{n} \sum_{j=1}^{n} e^{-i\frac{2\pi k}{n}j} e^{i\frac{2\pi k'}{n}(j+1)} = \sum_k \sum_{k'} b_k^\dagger b_{k'} e^{i\frac{2\pi k'}{n}} \frac{1}{n} \sum_{j=1}^{n} e^{i\frac{2\pi j}{n}(k'-k)} = \tag{31}$$

$$= \sum_k \sum_{k'} b_k^\dagger b_{k'} e^{i\frac{2\pi k'}{n}} \delta_{k,k'} = \sum_k b_k^\dagger b_k e^{i\frac{2\pi k}{n}},$$

$$\sum_{j=1}^{n} c_{j+1} c_j = \sum_{j=1}^{n} \left( \frac{1}{\sqrt{n}} \sum_k b_k e^{i\frac{2\pi k}{n}(j+1)} \right) \left( \frac{1}{\sqrt{n}} \sum_{k'} b_{k'} e^{i\frac{2\pi k'}{n}j} \right) =$$

$$= \sum_k \sum_{k'} b_k b_{k'} e^{i\frac{2\pi k}{n}} \frac{1}{n} \sum_{j=1}^{n} e^{i\frac{2\pi j}{n}(k'+k)} = \sum_k \sum_{k'} b_k b_{k'} e^{i\frac{2\pi k}{n}} \delta_{k,-k'} =$$

$$= \frac{1}{2} \left( \sum_{k'} b_{-k'} b_{k'} e^{-i\frac{2\pi k'}{n}} + \sum_k b_k b_{-k} e^{i\frac{2\pi k}{n}} \right) = \tag{32}$$

$$= \frac{1}{2} \left( \sum_k -b_k b_{-k} e^{-i\frac{2\pi k}{n}} + \sum_k b_k b_{-k} e^{i\frac{2\pi k}{n}} \right) = \sum_k i \sin\left( \frac{2\pi k}{n} \right) b_k b_{-k},$$

$$\sum_{j=1}^{n} c_j^\dagger c_{j+1}^\dagger = \sum_{j=1}^{n} \left( \frac{1}{\sqrt{n}} \sum_k b_k^\dagger e^{-i\frac{2\pi k}{n}j} \right) \left( \frac{1}{\sqrt{n}} \sum_{k'} b_{k'}^\dagger e^{-i\frac{2\pi k'}{n}(j+1)} \right) =$$

$$= \sum_k \sum_{k'} b_k^\dagger b_{k'}^\dagger e^{-i\frac{2\pi k'}{n}} \frac{1}{n} \sum_{j=1}^{n} e^{i\frac{2\pi j}{n}(-k'-k)} = \sum_k \sum_{k'} b_k^\dagger b_{k'}^\dagger e^{-i\frac{2\pi k'}{n}} \delta_{-k,k'} = \tag{33}$$

$$= \frac{1}{2} \left( \sum_{k'} b_{-k'}^\dagger b_{k'}^\dagger e^{-i\frac{2\pi k'}{n}} + \sum_k b_k^\dagger b_{-k}^\dagger e^{i\frac{2\pi k}{n}} \right) = \sum_k i \sin\left( \frac{2\pi k}{n} \right) b_k^\dagger b_{-k}^\dagger,$$

$$\sum_{j=1}^{n} c_{j+1}^\dagger c_j = \sum_{j=1}^{N} \left( \frac{1}{\sqrt{n}} \sum_k b_k^\dagger e^{-i\frac{2\pi k}{n}(j+1)} \right) \left( \frac{1}{\sqrt{n}} \sum_{k'} b_{k'} e^{i\frac{2\pi k'}{n}j} \right) =$$

$$\sum_k \sum_{k'} b_k^\dagger b_{k'} e^{-i\frac{2\pi k}{n}} \frac{1}{n} \sum_{j=1}^{n} e^{i\frac{2\pi j}{n}(k'-k)} = \tag{34}$$

$$= \sum_k \sum_{k'} b_k^\dagger b_{k'} e^{-i\frac{2\pi k}{n}} \delta_{k,k'} = \sum_k b_k^\dagger b_k e^{-i\frac{2\pi k}{n}},$$

$$\sum_{j=1}^{n} c_j^\dagger c_j = \sum_{j=1}^{N} \left( \frac{1}{\sqrt{n}} \sum_k b_k^\dagger e^{-i\frac{2\pi k}{n}(j)} \right) \left( \frac{1}{\sqrt{n}} \sum_{k'} b_{k'} e^{i\frac{2\pi k'}{n}j} \right) =$$

$$= \sum_k \sum_{k'} b_k^\dagger b_{k'} \frac{1}{n} \sum_{j=1}^{n} e^{i\frac{2\pi j}{n}(k'-k)} = \tag{35}$$

$$= \sum_k \sum_{k'} b_k^\dagger b_{k'} \delta_{k,k'} = \sum_k b_k^\dagger b_k.$$

Substituting the previous expressions in Eq.(25), one gets the transformed Hamiltonian

$$\mathcal{H}_{FT} = \sum_k \left[ 2 \left( \lambda + J \cos \left( \frac{2\pi k}{n} \right) \right) b_k^\dagger b_k + iJ\gamma \sin \left( \frac{2\pi k}{n} \right) (b_k^\dagger b_{-k}^\dagger + b_k b_{-k}) \right] - \lambda n. \tag{36}$$

As a result of working in momentum space, the XY-Hamiltonian does not contain mixed terms between first neighbors, however, it is not diagonal yet because it contains terms with opposite momentum $k$ and $-k$ coupled.

For future calculations, it is beneficial to rewrite the Eq.(36) making use of the cosine function parity $(\cos(\alpha) = \cos(-\alpha))$, acknowledging that the summation takes over positive and negative $k$ values and without carrying the constant term $\lambda n$. Thereafter, the Hamiltonian is expressed as follows

$$\mathcal{H}'_{FT} = \sum_k \left[ \left( \lambda + J \cos \left( \frac{2\pi k}{n} \right) \right) \left( b_k^\dagger b_k + b_{-k}^\dagger b_{-k} \right) + iJ\gamma \sin \left( \frac{2\pi k}{n} \right) (b_k^\dagger b_{-k}^\dagger + b_k b_{-k}) \right] =$$
$$= \sum_k \left[ \epsilon_k \left( b_k^\dagger b_k + b_{-k}^\dagger b_{-k} \right) + i\Delta_k (b_k^\dagger b_{-k}^\dagger + b_k b_{-k}) \right] = \tag{37}$$
$$= \sum_k \left[ \epsilon_k \left( b_k^\dagger b_k - b_{-k} b_{-k}^\dagger + 1 \right) + i\Delta_k (b_k^\dagger b_{-k}^\dagger + b_k b_{-k}) \right].$$

The last term can be rewritten in matrix-vector form, then the expression becomes

$$\sum_k \begin{pmatrix} b_k^\dagger & b_{-k} \end{pmatrix} \begin{pmatrix} \epsilon_k & i\Delta_k \\ -i\Delta_k & -\epsilon_k \end{pmatrix} \begin{pmatrix} b_k \\ b_{-k}^\dagger \end{pmatrix} + \sum_k \epsilon_k. \tag{38}$$

Here, the definitions of $\epsilon_k = \lambda + J \cos \left( \frac{2\pi k}{n} \right)$ and $\Delta_k = J\gamma \sin \left( \frac{2\pi k}{n} \right)$ serve the purpose of enhancing the clarity of the upcoming mathematical development.

## 2.3   Bogoliubov Transformation

The last step to diagonalize the Hamiltonian completely is the Bogoliubov transformation. This transformation is used to diagonalize quadratic Hamiltonians, for instance, it is used in the Superconductivity BSC theory or solid-state physics in Hamiltonians described by phononic interactions [19]. It can be understood as a change of basis, where the new base decouples the opposite momentum terms.

Specifically, the transformation will have a form such as

$$a_k = u_k b_k + v_k b_{-k}^\dagger, \qquad a_{-k} = u_{-k} b_{-k} + v_{-k} b_k^\dagger,$$
$$a_k^\dagger = u_k^* b_k^\dagger + v_k^* b_{-k}, \qquad a_{-k}^\dagger = u_{-k}^* b_{-k}^\dagger + v_{-k}^* b_k, \tag{39}$$

where $a_k^\dagger$ and $a_k$ are the Bogoulibov fermionic annihilation and creation operators associated with pseudo-momentum $k$, while $a_{-k}^\dagger$ and $a_{-k}$ are the Bogoulibov fermionic annihilation and creation operators associated with pseudo-momentum $-k$.

Because we are working in a fermionic system, we have to impose the anticommutation relationship of these new operators

$$\{a_k, a_k^\dagger\} = 1 \;\rightarrow\; |u_k|^2 + |v_k|^2 = 1,$$
$$\{a_k, a_{-k}\} = 0 \;\rightarrow\; u_k v_{-k} + v_k u_{-k} = 0 \tag{40}$$

To fulfill the second relationship, we use the condition $v_{-k} = -v_k$. This last condition along with Eq.(39) could be used to reverse the fermionic operator transformation. The old fermionic operators as a linear combination of the new fermionic operators are

$$b_k = u_k^* a_k - v_k a_{-k}^\dagger, \qquad b_{-k} = u_k^* a_{-k} + v_k a_k^\dagger,$$
$$b_k^\dagger = u_k a_k^\dagger - v_k^* a_{-k}, \qquad b_{-k}^\dagger = u_k a_{-k}^\dagger + v_k^* a_k. \tag{41}$$

For our purposes, it is useful to arrange the last expression in the vector-matrix form

$$\begin{pmatrix} b_k \\ b_{-k}^\dagger \end{pmatrix} = \begin{pmatrix} u_k^* & -v_k \\ v_k^* & u_k \end{pmatrix} \begin{pmatrix} a_k \\ a_{-k}^\dagger \end{pmatrix}. \tag{42}$$

The next step consists of passing from a non-diagonal Hamiltonian $\mathcal{H}_{FT}$ to a diagonal one by applying a change of basis matrix, which transforms the $b_k$ to $a_k$ operators.

$$\mathcal{H}'_{Bog} = \sum_k \begin{pmatrix} a_k^\dagger & a_{-k} \end{pmatrix} \begin{pmatrix} u_k & v_k \\ -v_k^* & u_k^* \end{pmatrix} \begin{pmatrix} \epsilon_k & i\Delta_k \\ -i\Delta_k & -\epsilon_k \end{pmatrix} \begin{pmatrix} u_k^* & -v_k \\ v_k^* & u_k \end{pmatrix} \begin{pmatrix} a_k \\ a_{-k}^\dagger \end{pmatrix}. \tag{43}$$

The Hamiltonian matrix written in terms of $a_k$ operators becomes

$$\begin{pmatrix} \epsilon_k \left(|u_k|^2 - |v_k|^2\right) + i\Delta_k \left(u_k v_k^* - u_k^* v_k\right) & -2\epsilon_k u_k v_k + i\Delta_k \left(u_k u_k + v_k v_k\right) \\ -2\epsilon_k u_k^* v_k^* - i\Delta_k \left(u_k^* u_k^* + v_k^* v_k^*\right) & -\left(\epsilon_k \left(|u_k|^2 - |v_k|^2\right) + i\Delta_k \left(u_k v_k^* - u_k^* v_k\right)\right) \end{pmatrix} \tag{44}$$

The Bogoliubov modes that diagonalize the Hamiltonian are found by vanishing the non-diagonal terms. For this purpose, it is convenient to express $u_k$ and $v_k$ as

$$u_k = e^{\phi_1} \cos\left(\frac{\theta_k}{2}\right), \qquad v_k = e^{\phi_2} \sin\left(\frac{\theta_k}{2}\right). \tag{45}$$

Substituting the last expression in the non-diagonal term of Eq.(44) and making it vanish, one gets the expression

$$- 2\epsilon_k e^{\phi_1 + \phi_2} \cos\left(\frac{\theta_k}{2}\right) \sin\left(\frac{\theta_k}{2}\right) + i\Delta_k u_k u_k - (-i\Delta_k) v_k v_k = 0 \rightarrow$$

$$- 2\epsilon_k e^{\phi_1 + \phi_2} \cos\left(\frac{\theta_k}{2}\right) \sin\left(\frac{\theta_k}{2}\right) + \Delta_k e^{2\phi_1 + \frac{\pi}{2}} \cos^2\left(\frac{\theta_k}{2}\right) - \Delta_k e^{2\phi_2 - \frac{\pi}{2}} \sin^2\left(\frac{\theta_k}{2}\right) = 0. \tag{46}$$

If one wishes to vanish the phase term in the expression, the relation $\phi_1 + \phi_2 = 2\phi_1 + \frac{\pi}{2} = 2\phi_2 - \frac{\pi}{2}$ must be fulfilled. Without loss of generality, the relative phase can be chosen as $\phi_1 = 0$ and $\phi_1 = \frac{\pi}{2}$. Accordingly, the new fermionic operators $a_k^\dagger$ and $a_k$ are

$$a_k = \cos\left(\frac{\theta_k}{2}\right) b_k + i\sin\left(\frac{\theta_k}{2}\right) b_{-k}^\dagger, \qquad a_{-k} = \cos\left(\frac{\theta_k}{2}\right) b_{-k} - i\sin\left(\frac{\theta_k}{2}\right) b_k^\dagger,$$

$$a_k^\dagger = \cos\left(\frac{\theta_k}{2}\right) b_k^\dagger - i\sin\left(\frac{\theta_k}{2}\right) b_{-k}, \qquad a_{-k}^\dagger = \cos\left(\frac{\theta_k}{2}\right) b_{-k}^\dagger + i\sin\left(\frac{\theta_k}{2}\right) b_k. \tag{47}$$

In addition, using the expressions $\sin(2\theta) = 2\cos(\theta)\sin(\theta)$ and $\cos(2\theta) = \cos^2(\theta) - \sin^2(\theta)$, the Eq.(46) becomes

$$\tan\left(\theta_k\right) = \frac{\Delta_k}{\epsilon_k}. \tag{48}$$

It is now possible to obtain the required expressions to compute the diagonal energy terms $(E_k)$

$$|u_k|^2 - |v_k|^2 = \cos^2\left(\frac{\theta_k}{2}\right) - \sin^2\left(\frac{\theta_k}{2}\right) = \cos\left(\theta_k\right) = \frac{1}{\sqrt{1 + \tan\left(\theta_k\right)}} = \frac{\epsilon_k}{\sqrt{\epsilon_k^2 + \Delta_k^2}},$$

$$u_k v_k = u_k^* v_k = \frac{i}{2}\sin\left(\theta_k\right) = \frac{i}{2}\tan\left(\theta_k\right)\cos\left(\theta_k\right) = \frac{i}{2}\frac{\Delta_k}{\sqrt{\epsilon_k^2 + \Delta_k^2}},$$

$$u_k v_k^* = u_k^* v_k^* = -\frac{i}{2}\frac{\Delta_k}{\sqrt{\epsilon_k^2 + \Delta_k^2}}, \tag{49}$$

$$E_k = \sqrt{\epsilon_k^2 + \Delta_k^2}.$$

Therefore, Eq.(43) has the diagonal form

$$\mathcal{H}'_{Bog} = \sum_k \begin{pmatrix} a_k^\dagger & a_{-k} \end{pmatrix} \begin{pmatrix} E_k & 0 \\ 0 & -E_k \end{pmatrix} \begin{pmatrix} a_k \\ a_{-k}^\dagger \end{pmatrix} = \sum_k E_k a_k^\dagger a_k - E_k a_{-k} a_{-k}^\dagger =$$

$$= \sum_k E_k \left(a_k^\dagger a_k + a_{-k}^\dagger a_{-k} - 1\right) = \sum_{k=\frac{-n}{2}+1}^{\frac{n}{2}} 2E_k \left(a_k^\dagger a_k - \frac{1}{2}\right). \tag{50}$$

Finally, the diagonal Hamiltonian has the form

$$\tilde{\mathcal{H}} = \sum_{k=\frac{-n}{2}+1}^{\frac{n}{2}} \left[2E_k \left(a_k^\dagger a_k - \frac{1}{2}\right) + \epsilon_k - \lambda\right], \tag{51}$$

where $E_k = \sqrt{\left(\lambda + J\cos\left(\frac{2\pi k}{n}\right)\right)^2 + \left(J\gamma\sin\left(\frac{2\pi k}{n}\right)\right)^2}$ are the energies related to having one fermion in the Bogoulibov mode $k$ or $-k$. As a result, we have diagonalized the XY Hamiltonian.

# 3    Quantum circuit to diagonalize the XY model

In this section, we introduce a circuit $\mathcal{U}_{\text{dis}}$ designed to convert the XY Hamiltonian $\mathcal{H}_{\text{XY}}$ into a non-interacting form $\bar{\mathcal{H}}_{\text{XY}}$, by

$$\bar{\mathcal{H}}_{\text{XY}} = \mathcal{U}_{\text{dis}}^{\dagger} \mathcal{H}_{\text{XY}} \mathcal{U}_{\text{dis}}. \tag{52}$$

Using this transformation, we can obtain all eigenstates and any superposition of them in the spin basis, by preparing a product state in the computational basis and applying $\mathcal{U}_{\text{dis}}^{\dagger}$,

$$|XY \text{ eigenstate}\rangle = \mathcal{U}_{\text{dis}}^{\dagger} |\text{Comp. basis}\rangle. \tag{53}$$

Furthermore, we can reverse this process. Applying $\mathcal{U}_{\text{dis}}$ to states in the computational basis allows us to obtain any spin state represented in the diagonal basis.

Unfortunately, constructing these disentangling circuits $\mathcal{U}_{dis}$ for an arbitrary Hamiltonian is a challenging task. However, for models that present analytical solutions, we can try to map each step into a quantum operation. For the case it concerns us, the XY Hamiltonian needs three operations: $i$) Jordan-Wigner transformation, $ii$) Fourier transform, $iii$) Bogoliubov transformation. In the end, the disentangling circuit will exhibit the structure

$$\mathcal{U}_{dis} = \mathcal{U}_{Bog}.\mathcal{U}_{FT}\mathcal{U}_{JW}. \tag{54}$$

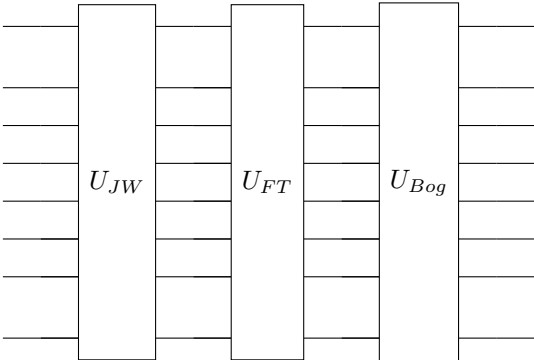

Figure 2: Schematic representation of the disentangling quantum circuit $U_{dis}$ for $n = 8$ qubits.

In the following section, we detail the construction of each $\mathcal{U}_{dis}$ operation using basic quantum gates.

## 3.1    Jordan-Wigner circuit

The Jordan-Wigner transformation maps the spin states to a fermionic spinless mode. In terms of the wave function,

$$\begin{aligned}
|\Psi\rangle &= \sum_{i_1,...,i_n=0,1} \Psi_{i_1,...,i_n} |i_1, ..., i_n\rangle = \\
&= \sum_{i_1,...,i_n=0,1} \Psi_{i_1,...,i_n} \left(c_1^{\dagger}\right)^{i_1} \cdots \left(c_n^{\dagger}\right)^{i_n} |0\rangle,
\end{aligned} \tag{55}$$

where $i_j$ represent the state $i$ of the qubit at position $j$, with $j$ going from 1 to the number of qubits $n$. In spin and fermionic space, the $i$ can take values 0 or 1. In spin space, $|0\rangle = |+\rangle$ and $|1\rangle = |-\rangle$, while in fermionic space $|0\rangle_j$ means the $j$-th position is not occupied by a fermion, and $|1\rangle_j$ means having one fermion.

The relevant point to observe is that the coefficients $\Psi_{i_1,...,i_n}$ do not change after the transformation. Therefore, theoretically, no gate is necessary to perform the Jordan-Wigner transformation. However, two subtleties are worth commenting on.

The first one becomes relevant when two-qubit states are exchanged through a SWAP operation. Since we are working with fermions, introducing a minus sign is necessary when two states are occupied

by fermions, such as $|11\rangle$. This adjustment is achieved through the fermionic SWAP operation (fSWAP). In matrix representation, the fSWAP is expressed as

$$fSWAP = \begin{pmatrix} 1 & 0 & 0 & 0 \\ 0 & 0 & 1 & 0 \\ 0 & 1 & 0 & 0 \\ 0 & 0 & 0 & -1 \end{pmatrix}, \tag{56}$$

which can be decomposed into a standard SWAP gate followed by a controlled-Z gate.

The second issue pertains to a notational problem. In conventional terms, spin states are denoted $|+\rangle = |0\rangle$, while in n-body systems the void state is written as $|0\rangle$ or sometimes $|\Omega\rangle$. Because Jordan-Wigner maps $|-\rangle$ into $|\Omega\rangle$, a decision has been made to introduce an $X$ gate. This choice preserves vector notation and mitigates confusion. Consequently, the initial circuit ensemble will consist of a layer of $X$ gates applied to each qubit. It is important to note this decision is primarily for adherence to conventions. Opting not to apply $X$ gates is a valid alternative. In such a case, the unitary transformation required to disentangle the $XY$ model will be slightly different from the description provided in this work, although the final result should remain unchanged.

## 3.2 Fermionic Fourier transform circuit

The next step is to get the fermionic modes to momentum space by applying the Fourier transform. For cases where the number of particles is a power of two, meaning $n = 2^m$ where $m$ is a natural number, the fermionic Fourier transform can be implemented by following the scheme of the classical Fast Fourier Transform [20].

The idea is based on the work of Andrew J. Ferrys in Ref.[21]. First, we decompose the n-qubit Fourier transform in two parallel $\frac{n}{2}$-qubit Fourier transforms, one acting upon odd and even modes respectively

$$\begin{aligned} b_k^\dagger &= \frac{1}{\sqrt{n}} \sum_{j=0}^{n-1} e^{i\frac{2\pi}{n}jk} c_j^\dagger = \frac{1}{\sqrt{\frac{n}{2}}2} \sum_{j'=0}^{n/2-1} e^{i\frac{2\pi}{n}2j'k} c_{2j'}^\dagger + \frac{1}{\sqrt{\frac{n}{2}}2} \sum_{j'=0}^{n/2-1} e^{i\frac{2\pi}{n}(2j'+1)k} c_{2j'+1}^\dagger = \\ &= \frac{1}{\sqrt{2}} \left[ \frac{1}{\sqrt{\frac{n}{2}}} \sum_{j'=0}^{n/2-1} e^{i\frac{2\pi}{n/2}j'k} c_{2j'}^\dagger + e^{i\frac{2\pi}{n}k} \frac{1}{\sqrt{\frac{n}{2}}} \sum_{j'=0}^{n/2-1} e^{i\frac{2\pi}{n/2}j'k} c_{2j'+1}^\dagger \right]. \end{aligned} \tag{57}$$

To avoid confusion between the operators defined previously we opted to use the tilde symbol for the operators obtained from the FT in this section. In this context, $b_k^\dagger$ is equivalent to the operator $b_k^\dagger$ defined in Sec.2.2.

We can now define a new set of fermionic operators for even and odd sites $a_j \equiv c_{2j'}$ and $d_j \equiv c_{2j'+1}$. The fermionic Fourier Transform of those operators using $\frac{n}{2}$ points will be

$$\begin{aligned} \tilde{a}_k^\dagger &= \frac{1}{\sqrt{\frac{n}{2}}} \sum_{j=0}^{\frac{n}{2}-1} e^{i\frac{2\pi}{n}jk} a_j^\dagger, \\ \tilde{d}_k^\dagger &= \frac{1}{\sqrt{\frac{n}{2}}} \sum_{j=0}^{\frac{n}{2}-1} e^{i\frac{2\pi}{n}jk} d_j^\dagger. \end{aligned} \tag{58}$$

If we now insert the prior definition in Eq.(57)

$$\begin{aligned} b_k^\dagger &= \frac{1}{\sqrt{2}} \left[ \tilde{a}_k^\dagger + e^{i\frac{2\pi}{n}k} \tilde{d}_k^\dagger \right], \\ b_{k+\frac{n}{2}}^\dagger &= \frac{1}{\sqrt{2}} \left[ \tilde{a}_k^\dagger - e^{i\frac{2\pi}{n}k} \tilde{d}_k^\dagger \right], \end{aligned} \tag{59}$$

where in the last equality we have used the periodicity of the Fourier Transform. In the case of $\frac{n}{2}$ Fourier Transform the period for $k$ values is $\frac{n}{2}$, so $\tilde{a}_{k+\frac{n}{2}}^\dagger = \tilde{a}_k^\dagger$ and exactly the same for $\tilde{d}_k^\dagger$ operator.

Equation (59) shows us that we can obtain the values of the $n$ qubit Fourier Transform ($b_k$) from a $\frac{n}{2}$ ($a_k, d_k$) qubit Fourier Transform. In the case of systems with $n = 2^m$ qubits, this process of division can persist until the Fourier Transform can be executed using a 2-qubit gate. Notably, the 2-qubit Fourier Transform has the same expression as Eq.(59) with $k = 0$.

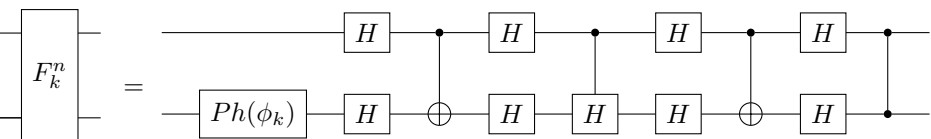

Figure 3: In the diagram is shown the decomposition of the building block of $F_k^n$ shown in Eq.(60), where $\phi_k = \frac{-i2\pi k}{n}$.

At this point, we have understood the interplay between $b_k$ and their counterparts $a_k$ and $d_k$. Nevertheless, our primary objective centers around obtaining the matrix that enables us to obtain the relationship between $|k\rangle_b |k + \frac{n}{2}\rangle_b$ and $|k\rangle_a |k\rangle_d$ states. This matrix can be obtained by acknowledging that the void state remains the same after the transformation, due to no mixing of annihilation and creation operators in the Fourier Transform definition. Hence, the remaining states can be attained by applying the creation operators to the void state, explicitly

- void vector (0 fermions) $\quad |0\rangle_{k_b} |0\rangle_{k_b+\frac{n}{2}} = |0\rangle_{k_a} |0\rangle_{k_d}$,

- apply $b_{k+\frac{n}{2}}^\dagger$ to obtain $|0\rangle_{k_b} |1\rangle_{k_b+\frac{n}{2}} = \frac{1}{\sqrt{2}} \left[ |1\rangle_{k_a} |0\rangle_{k_d} - e^{i\frac{2\pi}{n}k} |0\rangle_{k_a} |1\rangle_{k_d} \right]$,

- apply $b_k^\dagger$ to obtain $|1\rangle_{k_b} |0\rangle_{k_b+\frac{n}{2}} = \frac{1}{\sqrt{2}} \left[ |1\rangle_{k_a} |0\rangle_{k_d} + e^{i\frac{2\pi}{n}k} |0\rangle_{k_a} |1\rangle_{k_b} \right]$,

- apply $b_k^\dagger b_{k+\frac{n}{2}}^\dagger$ to obtain $|1\rangle_{k_b} |1\rangle_{k_b+\frac{n}{2}} = -e^{i\frac{2\pi}{n}k} |1\rangle_{k_a} |1\rangle_{k_d}$.

Here, the subscript $k_b$ means that this vector belongs to the $n$-qubit Fourier space, $k_a$ indicates that the vector is associated with the $n$ even-qubit Fourier space, and $k_d$ denotes that the vector belongs to the $n$ odd-qubit Fourier space.

Deriving the matrix that executes the operation described below is straightforward. Throughout the remainder of this work, we will refer to this matrix as the "General FT 2-qubit gate" or $F_k^n$. It takes the following form

$$
F_k^n \equiv \begin{pmatrix} 1 & 0 & 0 & 0 \\ 0 & \frac{-e^{-i\frac{2\pi}{n}k}}{2} & \frac{1}{\sqrt{2}} & 0 \\ 0 & \frac{e^{-i\frac{2\pi}{n}k}}{\sqrt{2}} & \frac{1}{\sqrt{2}} & 0 \\ 0 & 0 & 0 & -e^{-i\frac{2\pi}{n}k} \end{pmatrix}, \tag{60}
$$

where the $F_k^n$ matrix transforms the $|k\rangle_a |k\rangle_b$ vectors into $|k\rangle_c |k + \frac{n}{2}\rangle_c$. Additionally, the 2-qubit Fourier Transform is acquired whenever $k = 0$, which is represented as $F_2$.

It is key to bear in mind that there is a gap between the gates that can be applied theoretically in a quantum computer and the gates that we currently know how to implement in real devices. Therefore, all gates must be decomposed into basic gates that can be implemented in a quantum computer. For some cases, there are analytical schemes to decompose in basic gates [22]. For the case $F_k^n$, the basic gate decomposition is shown in Fig.3.

Up to this point, we have found the 2-qubit gate transform. Now we will describe the circuit scheme needed to perform the FT of $n$ qubits. The idea lies in decomposing our $n$ qubits into the even and odd sectors, and then applying the fermionic Fourier Transform of $\frac{n}{2}$ qubits. Once the $\frac{n}{2}$ Fourier Transform has been applied, the $F_k^n$ gate is applied to the $i$ qubit and the $i + \frac{n}{2}$ qubit. Finally, one has to repeat this scheme until all the Fourier Transforms are performed in 2 qubits, which will be performed by $F_2$. Nonetheless, assuming linear connectivity, where the qubits are arranged in a 1D line, additional fermionic SWAPS become necessary.

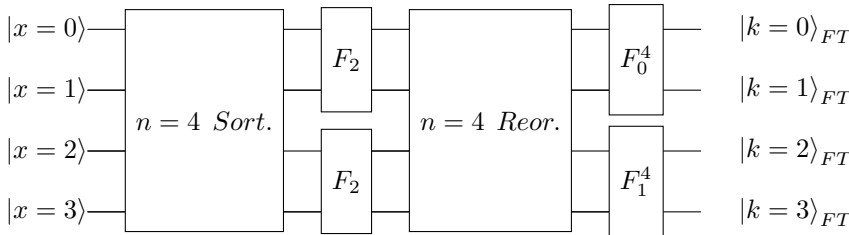

Figure 4: Scheme followed to perform the fermionic Fourier Transform (fFT) for the case of $n = 4$ qubits. The first step of the algorithm corresponds to the Qubit sorting (Sort.), then the fermionic Fourier Transform for $n = 2$ ($F_2$) qubits is applied and performed into the even and odd sectors. The next step is the Fourier states reorganization (Reor.) and finally, the General Fourier Transform 2-qubit gate ($F_k^n$) to recover the $k$ and $k + 2$ states.

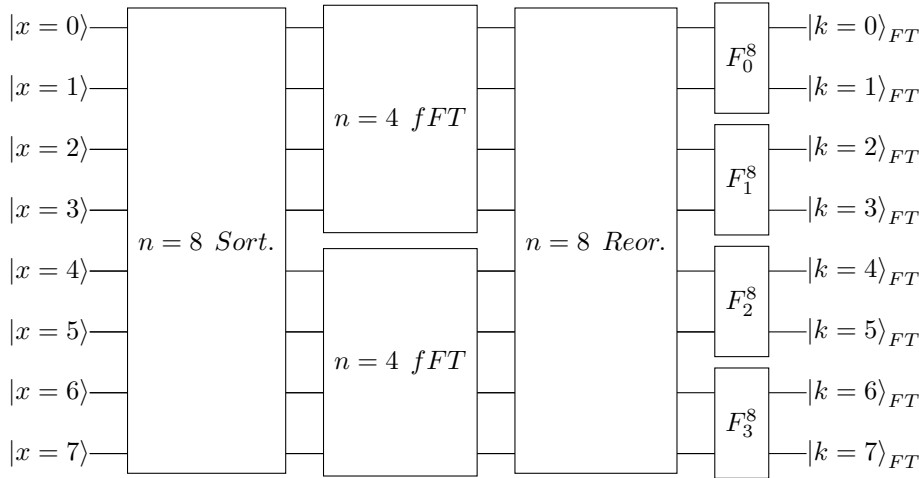

Figure 5: Scheme followed to perform the fermionic Fourier Transform (fFT) for the case of $n = 8$ qubits. The first step of the algorithm corresponds to the Qubit sorting (Sort.), then the fermionic Fourier Transform for $n = 4$ (fFT) qubits is applied and performed into the even and odd sectors. The next step is the Fourier states reorganization (Reor.) and finally the General Fourier Transform to recover the $k$ and $k + 4$ states.

Next, we will describe the algorithm used to construct the fermionic Fourier Transform circuit for $n$ qubits assuming linear connectivity and that the first qubit is numbered as 0. This circuit is decomposed into four phases:

1. **Qubit Sorting (Sort.):** In the initial step, we categorize the qubits into even and odd sectors using fermionic SWAP gates whenever we exchange two qubits.

2. $\frac{n}{2}$ **Fermionic Fourier Transform (fFT):** The second phase entails the application of the Fermionic Fourier Transform circuit for $\frac{n}{2}$ qubits into the even and odd sectors.

3. **Fourier states Reorganization (Reor.):** Subsequently, we undertake the reordering of the resulting states to group the $k_{even}$ and $k_{odd}$ states.

4. **General Fourier Transform Gate Application ($F_k^n$):** The final phase involves the application of the general Fourier transform gate $F_k^n$ to the $k_{even}$ and $k_{odd}$ states. This step is performed to recover the $k$ and $k + \frac{n}{2}$ states.

Figure 4 and Fig.5 represent the diagram of the fermionic Fourier Transform for the case of $n = 4$ and $n = 8$ qubits respectively. Both pictures show the different parts of the algorithm described above.

**Qubit Sorting**

The initial step involves the segregation of qubits into even and odd sectors through a series of fermionic SWAP operations, a process that occurs throughout $\frac{n}{2} - 1$ layers. In the first layer, precisely $\frac{n}{2} - 1$

fermionic gates come into play, each consecutively applied, starting with qubit 1. Subsequently, in each successive layer, one fewer gate is used than in the previous layer, following a sequential progression starting with the next qubit after the initial qubit of the preceding layer. Moreover, in Algorithm 1, we presented the algorithm in pseudocode:

---

**Algorithm 1** Qubit Sorting circuit

---

**Require:** $num\_qubit = 2^m$
**Ensure:** $qc\_sorting \rightarrow$ Quantum circuit which separates the qubits in even and odd sectors.
  $num\_label = \frac{n}{2} - 1$
  $num\_gates = \frac{n}{2} - 1$
  $qubit\_init = 1$
  **for** $i = 1$ to $num\_label$ **do**
    $count\_qubit = qubit\_init$
    **for** j=$num\_gates$ to 1 **do**
      add fSWAP into $count\_qubit$ and $count\_qubit + 1$
      $count\_qubit = count\_qubit + 2$
    **end for**
    $qubit\_init = qubit\_init + 1$
    $num\_gates = num\_gates - 1$
  **end for**

---

To enhance the accessibility and comprehensibility of the algorithm lecture, we have illustrated the circuit for the scenario where $n = 8$ in Fig.6. This visualization aims to make the algorithm more user-friendly and easier to use.

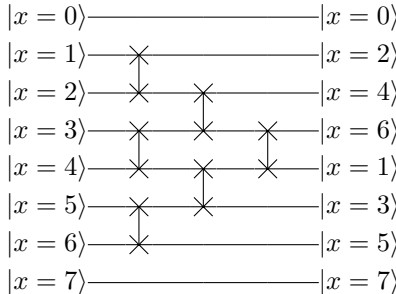

Figure 6: Qubit sorting circuit for the case of $n = 8$ qubits. Here, the fermionic SWAP gate has been represented using the same diagrammatic symbol as the SWAP gate.

## $\frac{n}{2}$ fermionic Fourier transform

The next step involves the application of two fermionic Fourier transforms to $\frac{n}{2}$ qubits, separately for the odd and even sectors. Consequently, the resulting vector states will correspond to momentum states labeled by $k$, ranging from $-\frac{n}{4} + 1$ to $\frac{n}{4}$ inclusive. However, it is important to highlight that the Fourier space is periodic, specifically in this case with a period of $\frac{n}{2}$. Consequently, it is possible to also label the $k$ states as 0 to $\frac{n}{2}$. In the case where $\frac{n}{2} = 2$, then the fermionic Fourier transform corresponds to applying $F_0^2$, which is described by Eq.(60). Figure 7 shows the scheme of the circuit for the $n = 8$ case.

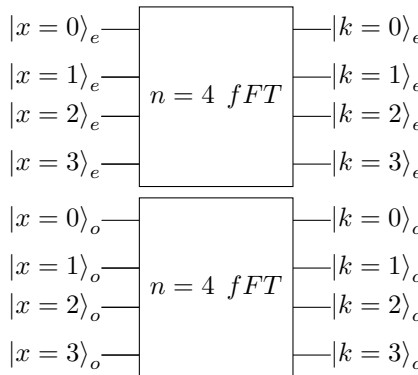

Figure 7: The $\frac{n}{2}$ fermionic Fourier transform circuit for the case of $n = 8$ qubits, the $e$ subindex stands for even while $o$ stands for odd. We use the periodicity of the Fourier transform, where the $k = 3$ state is equivalent to $k = -1$ state.

**Fourier states reorganization**

The reorganization phase aims to sort the new $\frac{n}{2}$ Fourier states groping the $k$ states from the even sector with the $k$ states from the odd sector. This is achieved by using the inverse circuit developed in the Qubit sorting step. Figure 7 shows the resulting circuit for the $n = 8$ case.

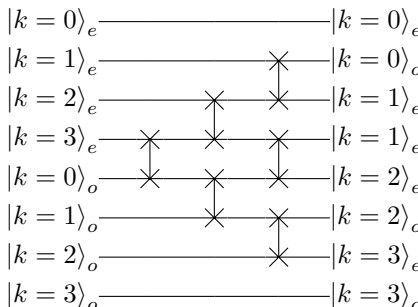

Figure 8: The Fourier states reorganization circuit for the case of $n = 8$ qubits, the $e$ subindex stands for even while $o$ stands for odd. Additionally, the SWAPs represented are fermionic SWAPs.

**General Fourier Transform Gate Application**

At this point, despite having obtained the $k$ states resulting from the $\frac{n}{2}$ fermionic Fourier transform, we still need to recover the $k$ states from the $n$ fermionic Fourier transform. To accomplish this last step, we have to apply the $F_k^n$ gate into the $|k_e\rangle$ and $|k_o\rangle$ states to recover the $|k\rangle$ and $|k + \frac{n}{2}\rangle$ states from the $n$ Fourier Transform.

---

**Algorithm 2** General Fourier Transform Gate Application circuit

**Require:** $num\_qubit = 2^m$
**Ensure:** $qc\_generalFT \rightarrow$ Quantum circuit which recovers the $n$ Fourier transform states $|k\rangle$ and $|k + \frac{n}{2}\rangle$ from the $\frac{n}{2}$ Fourier transform states $|k_e\rangle$ and $|k_o\rangle$.
  $num\_qubit = 0$
  **for** $k\_values= 0$ to $\frac{n}{2} - 1$ **do**
    Add the $F_k^n$ gate to qubit $num\_qubit$ and $num\_qubit + 1$ with $k = k\_values$.
    $num\_qubit = num\_qubit + 2$
  **end for**

---

We have illustrated the circuit for the scenario where $n = 8$ in Fig.9, to clarify the algorithm described.

Figure 9: The General Fourier Transform Gate Application circuit for the case of $n = 8$ qubits. Furthermore, we have illustrated the equivalence between the Fourier states, denoted by the $k$ labels ranging from $-\frac{n}{2} + 1$ to $\frac{n}{2}$ or from 0 to $n - 1$.

## 3.3 Bogoulibov transformation gate

The Bogoliubov transformation described in Eq.(47) mixes creation and annihilation operators from $k$ and $-k$ Fourier modes. Consequently, the void changed after being implemented the Bogoliubov transformation. The new void $|\Omega_0\rangle$ can be found in relation to the Fourier basis $|0\rangle, |k\rangle, |-k\rangle$ and $|-k, k\rangle$ imposing

$$
a_k |\Omega_0\rangle = \left( \cos\left(\frac{\theta_k}{2}\right) b_k + i \sin\left(\frac{\theta_k}{2}\right) b_{-k}^\dagger \right) \left( \delta |0\rangle + \alpha b_k^\dagger |0\rangle + \beta b_{-k}^\dagger |0\rangle + \gamma b_{-k}^\dagger b_k^\dagger |0\rangle \right) =
$$
$$
= \left( \cos\left(\frac{\theta_k}{2}\right) \alpha - \cos\left(\frac{\theta_k}{2}\right) \gamma b_{-k}^\dagger + i \sin\left(\frac{\theta_k}{2}\right) \delta b_{-k}^\dagger + i \sin\left(\frac{\theta_k}{2}\right) \alpha b_{-k}^\dagger b_k^\dagger \right) |0\rangle = 0,
$$

(61)

$$
a_{-k} |\Omega_0\rangle = \left( \cos\left(\frac{\theta_k}{2}\right) b_{-k} - i \sin\left(\frac{\theta_k}{2}\right) b_k^\dagger \right) \left( \delta |0\rangle + \alpha b_k^\dagger |0\rangle + \beta b_{-k}^\dagger |0\rangle + \gamma b_{-k}^\dagger b_k^\dagger |0\rangle \right) =
$$
$$
= \left( \cos\left(\frac{\theta_k}{2}\right) \beta + \cos\left(\frac{\theta_k}{2}\right) \gamma b_k^\dagger - i \sin\left(\frac{\theta_k}{2}\right) \delta b_k^\dagger - i \sin\left(\frac{\theta_k}{2}\right) \beta b_k^\dagger b_{-k}^\dagger \right) |0\rangle = 0.
$$

From the last equation, notice that $\alpha = 0$, $\beta = 0$ and $i \sin\left(\frac{\theta_k}{2}\right) \delta - \cos\left(\frac{\theta_k}{2}\right) \gamma = 0$. Using these constraints, the ground state is determined as

$$
|\Omega_0\rangle = \delta |0\rangle + \alpha |1_k\rangle + \beta |1_{-k}\rangle + \gamma |1_{-k} 1_k\rangle = \gamma \left( b_{-k}^\dagger b_k^\dagger + \frac{cos\left(\frac{\theta_k}{2}\right)}{i \sin\left(\frac{\theta_k}{2}\right)} \right) |0\rangle
$$

$$
\langle \Omega_0 | \Omega_0 \rangle = |\gamma|^2 \left( 1 + \frac{\cos^2\left(\frac{\theta_k}{2}\right)}{\sin^2\left(\frac{\theta_k}{2}\right)} \right)
$$

(62)

$$
|\gamma| = \sqrt{\frac{1}{1 + \frac{\cos^2\left(\frac{\theta_k}{2}\right)}{\sin^2\left(\frac{\theta_k}{2}\right)}}} = \sqrt{\sin^2\left(\frac{\theta_k}{2}\right)} = |\sin^2\left(\frac{\theta_k}{2}\right)|.
$$

Here, we have the freedom to choose the global phase of $\gamma$, we choose $\gamma = i \sin\left(\frac{\theta_k}{2}\right)$. Then the ground state is

$$
|\Omega_0\rangle = i \sin\left(\frac{\theta_k}{2}\right) |1_{-k} 1_k\rangle + cos\left(\frac{\theta_k}{2}\right) |0\rangle.
$$

(63)

Note that the new void vector only depends on the void of the FT and the $|1_k 1_{-k}\rangle$ Once the new void is acquired, the remaining vectors can be derived by applying the creation operators $a_k^\dagger$ and $a_{-k}^\dagger$

$$a_k^\dagger |\Omega_0\rangle = \left(\cos\left(\frac{\theta_k}{2}\right)b_k^\dagger - i\sin\left(\frac{\theta_k}{2}\right)b_{-k}\right)\left(cos\left(\frac{\theta_k}{2}\right)|0\rangle + i\sin\left(\frac{\theta_k}{2}\right)b_{-k}^\dagger b_k^\dagger |0\rangle\right) =$$

$$= \left(\cos^2\left(\frac{\theta_k}{2}\right)b_k^\dagger |0\rangle + 0\right) + \left(0 + \sin^2\left(\frac{\theta_k}{2}\right)b_k^\dagger |0\rangle\right) = |1_k\rangle\,,$$

$$a_{-k}^\dagger |\Omega_0\rangle = \left(\cos\left(\frac{\theta_k}{2}\right)b_{-k}^\dagger + i\sin\left(\frac{\theta_k}{2}\right)b_k\right)\left(cos\left(\frac{\theta_k}{2}\right)|0\rangle + i\sin\left(\frac{\theta_k}{2}\right)b_{-k}^\dagger b_k^\dagger |0\rangle\right) =$$

$$= \left(\cos^2\left(\frac{\theta_k}{2}\right)b_{-k}^\dagger |0\rangle + 0\right) + \left(0 - \sin^2\left(\frac{\theta_k}{2}\right)b_k b_{-k}^\dagger b_k^\dagger |0\rangle\right) = \tag{64}$$

$$= \cos^2\left(\frac{\theta_k}{2}\right)b_{-k}^\dagger |0\rangle + \sin^2\left(\frac{\theta_k}{2}\right)b_{-k}^\dagger |0\rangle = |1_{-k}\rangle\,,$$

$$a_{-k}^\dagger a_k^\dagger |\Omega_0\rangle = \left(\cos\left(\frac{\theta_k}{2}\right)b_{-k}^\dagger + i\sin\left(\frac{\theta_k}{2}\right)b_k\right)|1_k\rangle = \cos\left(\frac{\theta_k}{2}\right)|1_{-k}1_k\rangle + i\sin\left(\frac{\theta_k}{2}\right)|0\rangle\,.$$

Consider that the calculations have been done assuming the order $|-k, k\rangle$. However, for our purposes, it is more advantageous to rephrase this sequence as $|k, -k\rangle$, which entails incorporating a $-1$ whenever the state $|11\rangle$ is interchanged. The ultimate expressions are

$$|0_k 0_{-k}\rangle_{Bog} = cos\left(\frac{\theta_k}{2}\right)|0_k 0_{-k}\rangle_{FT} - i\sin\left(\frac{\theta_k}{2}\right)|1_k 1_{-k}\rangle_{FT}\,,$$

$$|1_k 0_{-k}\rangle_{Bog} = |1_k 0_{-k}\rangle_{FT}\,,$$

$$|0_k 1_{-k}\rangle_{Bog} = |0_k 1_{-k}\rangle_{FT}\,, \tag{65}$$

$$|1_k 1_{-k}\rangle_{Bog} = -i\sin\left(\frac{\theta_k}{2}\right)|0_k 0_{-k}\rangle_{FT} + \cos\left(\frac{\theta_k}{2}\right)|1_k 1_{-k}\rangle_{FT}\,,$$

where we have used a different notation. The $|k, -k\rangle_{Bog}$ corresponds to the Bogoulibov states, and The $|k, -k\rangle_{FT}$ corresponds to the Fourier states.

Once we have the Bogoulibov states written in terms of Fourier states, deriving the matrix that performs this operation is straightforward. Through the remainder of this work, we will refer to this matrix as the "Bogoulibov 2-qubit gate" or $B_k^n$. It takes the following form

$$B_k^n = \begin{pmatrix} \cos\left(\frac{\theta_k}{2}\right) & 0 & 0 & i\sin\left(\frac{\theta_k}{2}\right) \\ 0 & 1 & 0 & 0 \\ 0 & 0 & 1 & 0 \\ i\sin\left(\frac{\theta_k}{2}\right) & 0 & 0 & \cos\left(\frac{\theta_k}{2}\right) \end{pmatrix}\,, \tag{66}$$

where the $B_k^n$ matrix transforms the $|k, -k\rangle_{FT}$ vectors into $|k, -k\rangle_{Bog}$ and $\theta_k$ is described in Eq.(48).

The basic gate decomposition of $B_k^n$ is shown in Fig.10.

Figure 10: Decomposition of the building block of $B_k^n$ shown in Eq.(66), where $\theta_k$ is defined in Eq.(48).

The circuit scheme involves the decoupling of $k$ and $-k$ Fourier modes through the utilization of the two-qubit Bogoulibov gate denoted as $B_k^n$. While this task may seem straightforward, its complexity grows notably when we consider linear connectivity. This increased complexity emerges due to the necessity for additional fermionic SWAPS. These SWAPS are required to arrange the output states from the fermionic Fourier Transform, which initially groups the states $k$ and $k + \frac{n}{2}$, into the specific groupings of $k$ and $-k$ states needed for the Bogoulibov gates.

Next, we will describe the algorithm used to construct the Bogoulibov transformation circuit assuming linear connectivity and that the first qubit is numbered 0. This circuit is decomposed into two subcircuits:

1. **Bogoulibov Qubit Sorting:** The circuit consists of a series of fermionic SWAPS gates with the aim of grouping $k$ and $-k$ states.

2. **Bogoulibov Gate Application:** The circuit performs the Bogoulibov transformation applying the Bogoulibov gate into the modes $k$ and $-k$.

**Bogoulibov Qubit Sorting:**

The initial step involves segregating qubits into $k$ and $-k$ modes. This can be optimally achieved by employing $\frac{n}{4} - 1$ cascades of fermionic SWAP gates, arranged according to a specific geometric pattern.

The first cascade begins at qubit 3, followed by the next cascade, which starts at the succeeding qubit after the first four gates of the previous cascade have been applied. This sequencing is crucial for optimizing the circuit's depth. If the second cascade is initiated before the completion of the fourth gate in the previous one, it would result in an incorrect sorting of states. While there are other, more straightforward geometries that can be programmed, such as applying cascades sequentially, they do increase the overall circuit depth.

Each cascade initially consists of $n - 4$ consecutive fermionic SWAP gates, each starting where the previous one left off. Subsequently, an additional $n - 5$ fermionic SWAP gates are applied sequentially, with each gate being applied one level above the previous one.

In Algorithm 3, we presented the algorithm in pseudocode:

---

**Algorithm 3** Qubit Sorting circuit

---

**Require:** $num\_qubit = 2^m$
**Ensure:** $qc\_Bog\_sorting \rightarrow$ Quantum circuit which groups the $k$ and $-k$ Fourier states.
  $num\_cascade = \frac{n}{4} - 1$
  $qubit\_init = 3$
  **if** $num\_cascade = 2$ **then**
    stop
  **else**
    **for** $i = num\_cascade$ to 1 **do**
      $count\_qubit = qubit\_init$
      $down\_cascade = i \cdot 4$
      $up\_cascade = (i \cdot 4) - 1$
      **for** $j = 1$ to $down\_cascade$ **do**
        Add fSWAP into $count\_qubit$ and $count\_qubit + 1$
        $count\_qubit = count\_qubit + 1$
      **end for**
      **for** $j = 1$ to $up\_cascade$ **do**
        Add fSWAP into $count\_qubit - 1$ and $count\_qubit$
        $count\_qubit = count\_qubit - 1$
      **end for**
      $qubit\_init = qubit\_init + 1$
      Start after the $4th$ fSWAP of the previous cascade
    **end for**
  **end if**

---

To enhance the accessibility and comprehensibility of the algorithm lecture, we have illustrated the circuit for the scenario where $n = 8$ in Fig.11. This visualization aims to make the algorithm more user-friendly and easier to use.

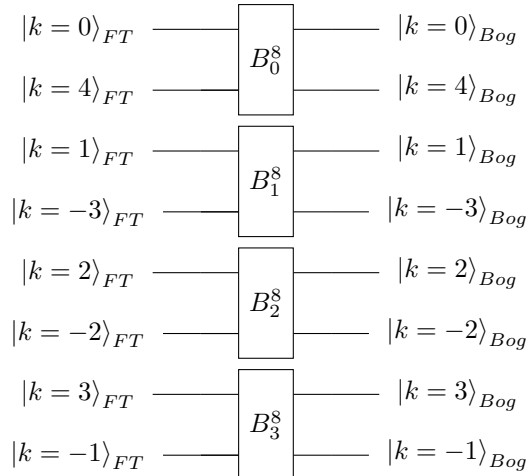

Figure 12: In the diagram is shown the Bogoulibov Gate Application circuit for the case of $n = 8$ qubits.

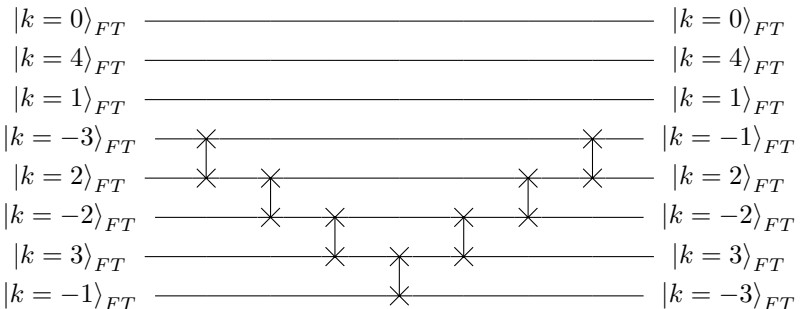

Figure 11: Bogoulibov qubit sorting circuit for the case of $n = 8$ qubits. Here, the fermionic SWAP gate has been represented using the same diagrammatic symbol as the SWAP gate.

**Bogoulibov Gate Application:**

Finally, we have arrived at the last step to obtain our diagonalizing circuit. To disentangle the $k$ and $-k$ states, the Bogoulibov gate $B_k^n$ is applied. Hence, the new circuit will be simply a layer of Bogoulibov gates, where each gate will act on the corresponding $k$ and $-k$ states, starting from $k = 0$ to $k = \frac{n}{2} - 1$.

---

**Algorithm 4** Bogoulibov Gate Application circuit

---

**Require:** $num\_qubit = 2^m$
**Ensure:** $qc\_generalBog \rightarrow$ Quantum circuit which disentangles the $|k\rangle$ and $|-k\rangle$.
  $num\_qubit = 0$
  **for** $k\_values = 0$ to $\frac{n}{2} - 1$ **do**
    Add the $B_k^n$ gate to qubit $num\_qubit$ and $num\_qubit + 1$ with $k = k\_values$.
    $num\_qubit = num\_qubit + 2$
  **end for**

---

We have illustrated the circuit for the scenario where $n = 8$ in Fig.12, where can be stated the similarity with the General Fourier Transform circuit.

## 3.4 Example: $n = 4$ and $n = 8$ spin chain

The explicit circuit $U_{dis}$ for spin chains with $n = 4$ and $n = 8$ is illustrated in Fig.13, Fig.14, and Fig.15. As an example of the many applications facilitated by $U_{dis}$, we performed simulations to evaluate the ground state and first excited state energies of the symmetric XY model ($J = 1$ and $\gamma = 0$) for spin chains of $n = 4$ and $n = 8$, considering various values of $\lambda$. Computing the energy of the ground and first excited state enables us to observe the quantum phase transition from an antiferromagnetic to a

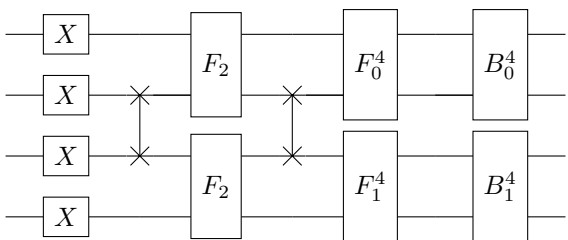

Figure 13: Quantum circuit $U_{dis}$ designed to diagonalize the XY Hamiltonian for $n = 4$ qubits. The initial layer consists of X gates, executing the Jordan-Wigner transformation. Subsequently, $F_2$ and $F_k^n$ implement the fermionic Fourier Transform. The circuit concludes with the Bogoliubov transformation achieved by $B_k^n$. Additionally, the swaps represented in the diagram correspond to fermionic SWAPs.

paramagnetic state, as mentioned in Sec.2. In the symmetric XY model, the diagonalized Hamiltonian becomes

$$\mathcal{H} = \sum_{k=\frac{-n}{2}+1}^{\frac{n}{2}} 2\left(\lambda + J\cos\left(\frac{2\pi k}{n}\right)\right) b_k^\dagger b_k - \lambda n, \tag{67}$$

where $b_k^\dagger b_k$ is the number occupation of the Fourier states $k$. Notice from Eq.(48), that in the case $\gamma = 0$ the Fourier and Bogoulibov modes are equivalents. For the case $n = 4$, the ground and the first excited state written in the diagonal basis are

$$|gs\rangle = \begin{cases} |0,1,0,0\rangle, & \lambda \le 1, \\ |0,0,0,0\rangle, & \lambda \ge 1, \end{cases} \qquad |e\rangle = \begin{cases} |0,0,0,0\rangle, & \lambda \le 1, \\ |0,1,0,0\rangle, & \lambda \ge 1. \end{cases} \tag{68}$$

For the case $n = 8$, the ground and the first excited state are

$$|gs\rangle = \begin{cases} |0,1,0,0,0,0,0,0\rangle, & \lambda \le 1, \\ |0,0,0,0,0,0,0,0\rangle, & \lambda > 1, \end{cases} \qquad |e\rangle = \begin{cases} |0,0,0,0,0,0,0,0\rangle, & \lambda \le 1, \\ |0,1,0,0,0,0,0,0\rangle, & \lambda > 1. \end{cases} \tag{69}$$

Additionally, we have simulated the ground state for the transverse field Ising model ($J = 1$ and $\gamma = 1$) in the $n = 4$ spin chain followed by the computation of the corresponding transverse magnetization. We have chosen magnetization because is one of the physical parameters which enable us to observe the phase transition discussed before. Analytically, the $\langle M_z \rangle = \sum_{i=1}^{n} \sigma_i^z$ in the ground state is

$$\langle gs|M_z|gs\rangle = \begin{cases} -\frac{\lambda}{2\sqrt{1+\lambda^2}}, & \lambda \le 1, \\ -\frac{1}{2} - \frac{\lambda}{2\sqrt{1+\lambda^2}}, & \lambda \ge 1. \end{cases} \tag{70}$$

# 4 Time evolution

We have presented the disentangling circuit $U_{dis}$ for the 1-D XY model, providing us with the capability to explore the complete Hamiltonian spectrum through a straightforward application of the disentangling circuit to the computational basis. This facilitates the calculation of diverse system properties, such as the expectation value of energy and magnetization, as detailed in the preceding section.

However, there are instances where our focus is on computing dynamic properties. In such cases, we need to calculate the time evolution of the state, which can be a challenging task. Nonetheless, a quantum circuit can be constructed to achieve exact time evolution for fermionic Hamiltonians that can be decomposed as the sum of the energies of each particle independently,

$$\mathcal{H} = \sum_{\alpha=1}^{N} \epsilon_\alpha a_\alpha^\dagger a_\alpha, \tag{71}$$

where $\epsilon_\alpha$ is the energy associated with having a particle in the state $\alpha$, $a_\alpha^\dagger$ and $a_\alpha$ are the fermionic creation and annihilation operator of the particle in the given state.

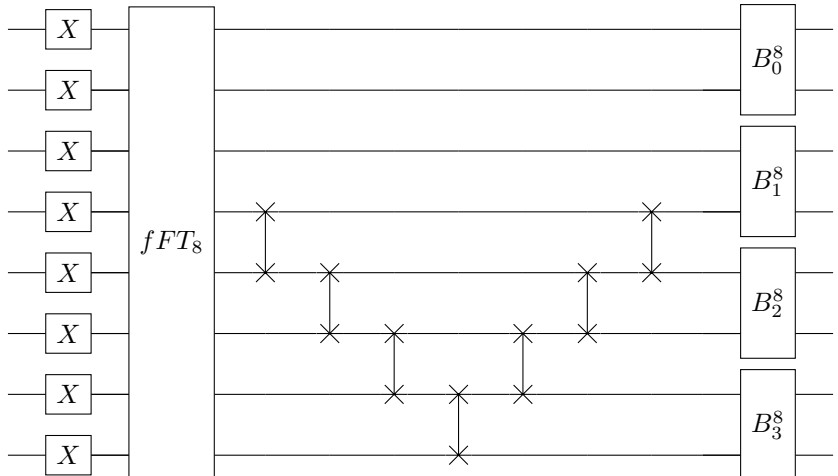

Figure 14: Quantum circuit $U_{dis}$ designed to diagonalize the XY Hamiltonian for $n = 8$ qubits. The initial layer consists of X gates, executing the Jordan-Wigner transformation. Subsequently, the fermionic Fourier Transform is applied by the circuit $fFT_8$, described in Fig.15. The circuit concludes with the Bogoliubov transformation achieved by $B_k^n$. Additionally, the swaps represented in the diagram correspond to fermionic SWAPs.

The reason behind constructing the time evolution gate in this case is straightforward, lies in the fact that for such Hamiltonians, the general time evolution operator $\mathcal{U}(t)$ can be decomposed into a product state of the time evolution operator for each qubit. To illustrate this, let's express the general time evolution of a given state $|\psi(t)\rangle$ driven by a non-time-dependent Hamiltonian $\mathcal{H}$. The evolution is accomplished by the unitary time-evolution operator $\mathcal{U}(t)$

$$\mathcal{U}(t) \equiv e^{-it\mathcal{H}},$$
$$|\psi(t)\rangle = \mathcal{U}(t)|\psi_0\rangle = \sum_l e^{-itE_l}|E_l\rangle\langle E_l||\psi_0\rangle, \tag{72}$$

where $|\psi_0\rangle$ is the initial state, $|E_l\rangle$ are the eigenstates of the given Hamiltonian, and $E_l$ are the corresponding energies or eigenvalues of each state $|E_l\rangle$.

Due to the decomposable form of Hamiltonian in Eq.(71), eigenstates can be expressed as a product state of $N$ states, each representing the presence or absence of a fermion in the $\alpha$ state

$$|E_l\rangle = |\alpha = 1\rangle|\alpha = 2\rangle \cdots |\alpha = N\rangle. \tag{73}$$

where $|\alpha\rangle$ can be represented by the qubits $|0\rangle$ or $|1\rangle$. Consequently, the time evolution operator becomes

$$\mathcal{U}(t)|E_l\rangle = e^{-it\mathcal{H}}|\alpha = 1\rangle|\alpha = 2\rangle \cdots |\alpha = N\rangle = e^{-it\sum_{\alpha=1}^N \epsilon_\alpha a_\alpha^\dagger a_\alpha}|\alpha = 1\rangle|\alpha = 2\rangle \cdots |\alpha = N\rangle =$$
$$= \mathcal{U}_1|\alpha = 1\rangle\mathcal{U}_2|\alpha = 2\rangle \cdots \mathcal{U}_n|\alpha = N\rangle, \tag{74}$$

where $\mathcal{U}_i$ is the time evolution operator for the $i_{th}$ qubit.

The procedure described above, tells us that to construct the time evolution circuit we just need to perform time evolution for each qubit independently. Specifically for the XY 1-D model, the time evolution for the qubit representing a fermionic particle with momentum $k$ is

$$\mathcal{U}_k = e^{-it2E_k a_k^\dagger a_k} e^{-it[-E_k + \epsilon_k - \lambda]} = U_1 U_2, \tag{75}$$

where $\epsilon_k = \lambda + J\cos(\frac{2\pi k}{n})$ and $E_k = \sqrt{\left(\lambda + J\cos\left(\frac{2\pi k}{n}\right)\right)^2 + \left(J\gamma\sin\left(\frac{2\pi k}{n}\right)\right)^2}$ are the energies associated to having one fermion in the Bogoulibov mode $k$.

The unitary operators $U_1$ and $U_2$ can be written in matrix form

$$U_1 = e^{-it2E_k a_k^\dagger a_k} = \begin{pmatrix} 1 & 0 \\ 0 & e^{-it2E_k} \end{pmatrix} = \begin{pmatrix} 1 & 0 \\ 0 & e^{i\varphi_k} \end{pmatrix},$$
$$U_2 = \begin{pmatrix} e^{-it[-E_k + \epsilon_k - \lambda]} & 0 \\ 0 & e^{-it[-E_k + \epsilon_k - \lambda]} \end{pmatrix} = \begin{pmatrix} e^{i2\Phi_k} & 0 \\ 0 & e^{i2\Phi_k} \end{pmatrix}, \tag{76}$$

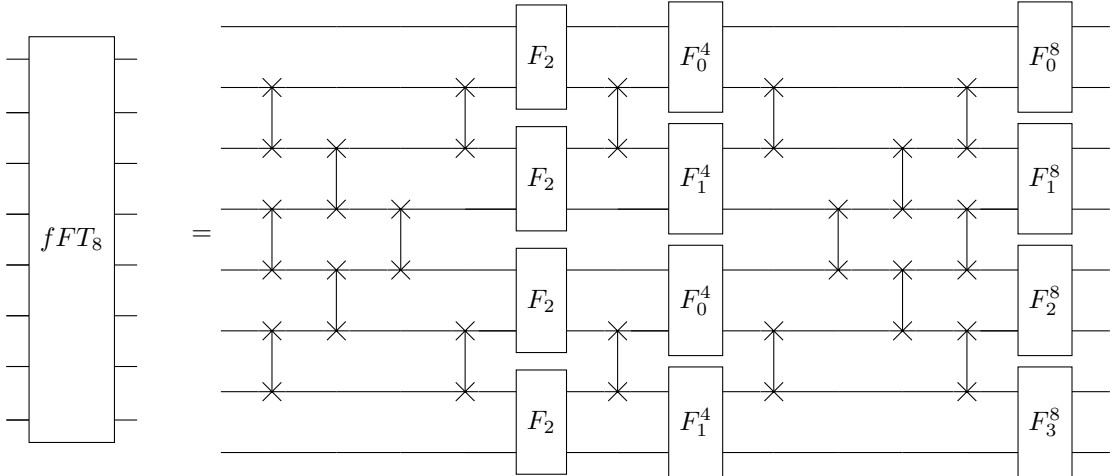

Figure 15: Fermionic Fourier Transform circuit for the case $n = 8$. The swaps represented in the diagram correspond to fermionic SWAPs.

whereby $E_k = \sqrt{\left(\lambda + J\cos\left(\frac{2\pi k}{n}\right)\right)^2 + \left(J\gamma\sin\left(\frac{2\pi k}{n}\right)\right)^2}$ and $\epsilon_k = \lambda + J\cos\left(\frac{2\pi k}{n}\right)$. Additionally, we have renamed the exponential arguments by $\varphi_k = -2tE_k$, and $\Phi_k = -2t\left[-E_k + \epsilon_k - \lambda\right]$. The gate decomposition of $\mathcal{U}_k$ is shown in Fig.16.

$$-\boxed{\mathcal{U}_k}- \quad = \quad -\!\!-\!\!-\boxed{Ph(\varphi_k)}-\boxed{Ph(\frac{\Phi_k}{2})}-\boxed{X}-\boxed{Ph(\frac{\Phi_k}{2})}-\boxed{X}-\!\!-$$

Figure 16: In the diagram is shown the decomposition of the building block of $\mathcal{U}_k$ shown in Eq.(76), where $\varphi_k = -2tE_k$, and $\Phi_k = -2t\left[-E_k + \epsilon_k - \lambda\right]$.

As an example of the many possibilities this gate opens, let's compute the time evolution of the expected value of transverse magnetization for the $n = 4$ qubits case, with $J = 1$ and $\gamma = 1$. Specifically, our initial state has all the spins aligned in the positive $z$ direction $|\psi(t = 0)\rangle = |\uparrow\;\uparrow\;\uparrow\;\uparrow\rangle$, which in the computational basis is written as $|0000\rangle$ state. The first step to compute the time evolution consists of expressing the initial state in the eigenbasis of the XY Hamiltonian. This is achieved by precisely applying the $U_{dis}$ gate

$$|\psi(t = 0)\rangle = \mathcal{U}_{dis}|0\;0\;0\;0\rangle = \sin\left(\frac{\phi}{2}\right)|1\;1\;0\;0\rangle - i\cos\left(\frac{\phi}{2}\right)|1\;1\;1\;1\rangle, \tag{77}$$

where $\phi = \operatorname{arctg}\left(\frac{1}{\lambda}\right)$. Subsequently, we apply the time evolution operator $U(t)$ to obtain $|\psi(t)\rangle$. Then, the time-dependent state is

$$|\psi(t)\rangle = e^{-it2\left(\lambda-\sqrt{1+\lambda^2}\right)}\sin\left(\frac{\phi}{2}\right)|1\;1\;0\;0\rangle - ie^{-it2\left(\lambda+\sqrt{1+\lambda^2}\right)}\cos\left(\frac{\phi}{2}\right)|1\;1\;1\;1\rangle =,$$

$$= e^{-it2\lambda}e^{-i2t\sqrt{1+\lambda^2}}\left(e^{+i4t\sqrt{1+\lambda^2}}\sin\left(\frac{\phi}{2}\right)|1\;1\;0\;0\rangle - i\cos\left(\frac{\phi}{2}\right)|1\;1\;1\;1\rangle\right), \tag{78}$$

where the global phases are not physically relevant. After applying the time operator, we now apply the circuit $U^\dagger$ to obtain the state in the spin representation. Lastly, we compute analytically the expected value of the transverse magnetization $\langle M_z \rangle$, which yields the analytical result

$$\langle M_z \rangle = \frac{1 + 2\lambda^2 + \cos\left(4t\sqrt{1+\lambda^2}\right)}{2 + 2\lambda^2}. \tag{79}$$

# 5 Results and discussion

In this section, we delve into the outcomes and insights derived from the application of our quantum circuit, $U_{dis}$, across various scenarios. All the simulations for the spin chain $n = 4$ and $n = 8$ have been performed using the circuits represented in Figs. 13, 14, and 15.

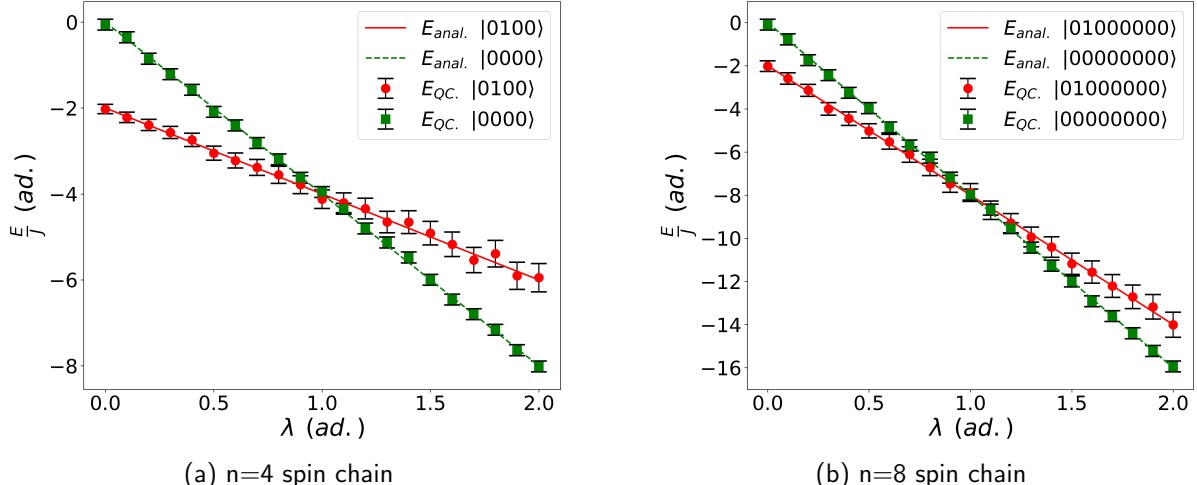

|               |               |
| :-----------: | :-----------: |
| (a) n=4 spin chain | (b) n=8 spin chain |

Figure 17: Study of the ground and first excited state energy for the symmetric XY model ($J = 1$ and $\gamma = 0$) as a function of the transverse field strength parameter $\lambda$. The solid (dashed) line represents the analytical values of the energy $E$, while the scatter points correspond to results obtained from a quantum computer simulation conducted in Qibo. (a) shows results for an $n = 4$ spin chain, and (b) for an $n = 8$ spin chain.

Figure 17 presents the outcomes of the expected energy for the ground and first excited states in the symmetric XY model ($J = 1$. and $\gamma = 0$) for spin chains with $n = 4$ and $n = 8$. Given the nature of quantum simulations, subject to inherent probabilistic uncertainties, each data point carries a statistical error proportional to $\frac{1}{\sqrt{N}}$, where $N$ represents the number of shots—indicating the executions on a quantum processing unit (QPU). Here, $N$ was set to 1000. Notably, the results showcase the circuit's effectiveness in recovering analytical values for both cases. Moreover, a structural change in the ground state is evident at $\lambda = 1$, where the more stable state becomes the one without particles in the Bogoliubov modes $k$ instead of having a fermion in the $-k$ mode.

For the transverse field Ising model ($J = 1$ and $\gamma = 1$) in the $n = 4$ spin chain, the results of the ground state's expected value of transverse magnetization $\langle M_z \rangle$ are shown in Fig. 18. The circuit successfully reproduces analytical values, and at $\lambda = 1$, a magnetization discontinuity occurs due to a phase transition from an antiferromagnetic state to a paramagnetic state.

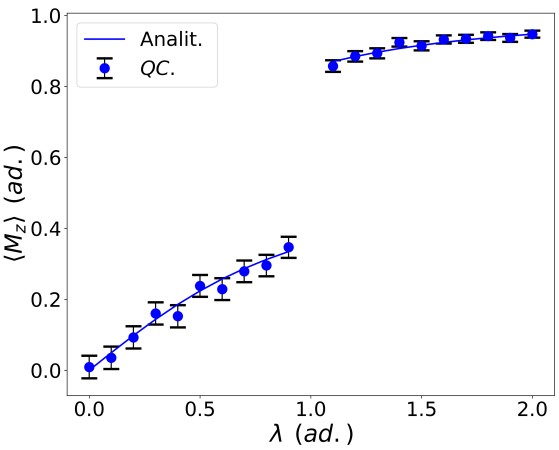

Figure 18: The ground state's expected value of transverse magnetization $\langle M_z \rangle$ for the transverse field Ising model ($J = 1$ and $\gamma = 1$) in a spin chain with $n = 4$ spins, as a function of the transverse field strength parameter $\lambda$. The solid line represents the analytical value of $\langle M_z \rangle$, while the scatter points correspond to the results obtained from a quantum computer simulation conducted in Qibo, utilizing the quantum circuit developed in this paper.

Moreover, we have also used the transverse field Ising model ($J = 1$ and $\gamma = 1$) to explore the time evolution of the expected value of transverse magnetization $\langle M_z(t) \rangle$. The quantum circuit $\mathcal{U}(t)$ is applied to evolve the initial state $|\uparrow, \uparrow, \uparrow, \uparrow\rangle$ with the magnetic field strength fixed at $\lambda = 0.5$. After, we apply $U_{dis}^\dagger$ to obtain the evolved spin state. The results are shown in Fig.19, showcasing successful agreement between the quantum simulation and analytical values.

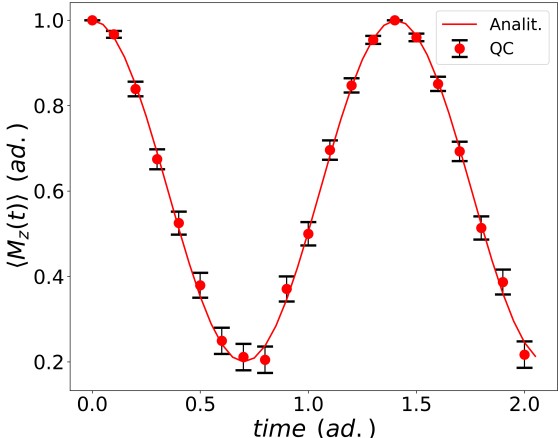

Figure 19: Time evolution simulation of transverse magnetization $\langle M_z \rangle$ for the transverse field Ising model ($J = 1$ and $\gamma = 1$) in a spin chain with $n = 4$ spins. The initial spin state is $|\uparrow, \uparrow, \uparrow, \uparrow\rangle$, evolved using the quantum circuit $\mathcal{U}(t)$ with the magnetic strength fixed at $\lambda = 0.5$. The solid line represents the analytical value of $\langle M_z \rangle$, while the scatter points correspond to the results obtained from a quantum computer simulation conducted in Qibo, utilizing the quantum circuit developed in this paper.

The circuit presented scales efficiently with the number of qubits. The Jordan-Wigner transformation is a simple layer of $X$ gate, as a result, escalates linearly with the number of qubits and the depth is constant. Similarly, the Bogoulibov transformation only combines $k$ and $-k$ modes, resulting in a constant circuit depth while the number of gates escalates proportionally to $\sim \frac{n}{2}$, where $n$ represents the number of qubits. In Ref.[21], it is shown that the circuit depth of the Fourier transforms follows a logarithmic scaling of $\sim \log_2(n)$, with the number of gates increasing as $\sim n \log_2(n)$. The time evolution circuit scales linearly with the number of qubits $n$ and presents a constant depth.

# 6    Conclusion

This paper presents a comprehensive implementation of the exact simulation of a 1-D XY spin chain using a digital quantum computer. Our approach encompasses the entire solution process for this exactly solvable model, involving key transformations such as the Jordan-Wigner transformation, fermionic Fourier transform, and Bogoliubov transformation. Additionally, we developed an algorithm to construct an efficient quantum circuit for powers of two qubits, capable of diagonalizing the XY Hamiltonian and executing its exact time evolution. The explicit code to reproduce these circuits is presented in Ref.[11] and uses Qibo, an open-source framework for quantum computing.

The presented quantum circuit is a powerful tool, facilitating the calculation of all eigenstate vectors by initializing qubits on a computational basis and subsequently applying the detailed circuit. This feature enables access to the complete spectrum of the Hamiltonian, providing novel approaches for exploring various system properties, including energy, magnetization, and time evolution.

Our introduced quantum circuit serves as a benchmark for quantum computing devices. It presents efficient growth and scalability with the number of qubits $n$, making it suitable to be used in devices of diverse sizes. Furthermore, the 1-D XY model's exact solvability not only allows us to test the efficiency of real quantum computers but it offers an avenue to study and model errors inherent in quantum computations, establishing a bridge between theoretical predictions and real-world outcomes.

Beyond its utility as a benchmark, the presented quantum circuit holds intriguing applications in condensed matter physics. The methods highlighted in this work can be extended to explore other integrable models, such as the Kitaev Honeycomb model [10], or with alternative ansatz, as seen in the Heisenberg model [12]. Additionally, there is potential to employ different strategies for simulating thermal evolution [8] and could open new paths to study quantum phase transition.

In conclusion, our work contributes to the advancement of quantum computing algorithms and establishes a foundation for exploring quantum solutions to complex problems in condensed matter physics.

# Acknowledgments

A. C.-L. acknowledges funding from the Spanish Ministry for Digital Transformation and of Civil Service of the Spanish Government through the QUANTUM ENIA project call - Quantum Spain, EU through the Recovery, Transformation and Resilience Plan – NextGenerationEU within the framework of the Digital Spain 2026.

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
