# Peer review of "Simulation of the 1d XY model on a quantum computer"

_SciPost Physics_

## Round 1 · Referee Report · Anonymous (Referee 2) · 2025-1-6

Report

The article presents a method for performing the exact diagonalization of the XY Hamiltonian through a dedicated quantum circuit. The text has a pedagogical character, with long sections dedicated to explaining, in full detail, well-known techniques (see the entire Section 2).

I believe the article can be published for its pedagogical value, as it could serve as an extremely useful reference for students approaching the topic. I believe publication in SciPost Lecture Notes could also be a viable option to consider. In any case, I would suggest a revision of the text, as it is not very well-written in some parts.

Here, I highlight some issues I found in the text, along with improved versions. However, I think there are many other issues throughout the text...

-"In this paper, we present the comprehensive scheme for..." -> "In this paper, we present a comprehensive scheme for..."

-"These transitions occur at absolute zero..." -> "These transitions occur at zero temperature..."

-"The quadratic Hamiltonian in fermionic annihilation and creation operators appears in more condensed matter systems notably exemplified in the Hubbard model" -> "Hamiltonians quadratic in fermionic annihilation and creation operators are ubiquitous in condensed matter systems, describing systems of free fermionic particles." Concerning the Hubbard model: usually this model is considered with the interaction term, so is not quadratic...

-"...leading us to the subsequent phase: the fermionic Fourier transform (fFT)." -> "Diagonalizing this type of Hamiltonian is a well-established process, achieved through the fermionic Fourier transform (fFT)."

-"...our primary objective centers around obtaining the matrix" -> "...our primary objective is to obtain the matrix"

-"Figure 3: In the diagram is shown the decomposition of the building block of Fnk shown in Eq.(60), where ϕk =−i2πkn." -> "Figure 3: The diagram illustrates the decomposition of the building block of Fnk (Eq.(60)), where ϕk =−i2πkn."

While I did not review all the equations in detail, I believe they are essentially correct.

Other questions:

-Why you say the circuit is "specifically designed for the NISQ era"? What is NISQ specific here?

-Shouldn't the order of the three unitaries in Eq. (54) be reversed? Based on Eq. (52), it seems that U_JW should be applied to H first...

-Could your circuit serve as a starting point for a variational quantum circuit designed to address more complex, interacting Hamiltonians?

Recommendation

Ask for major revision

  • validity: -
  • significance: -
  • originality: low
  • clarity: high
  • formatting: reasonable
  • grammar: acceptable

Author:  Marc Farreras  on 2025-03-11  [id 5276]

(in reply to Report 2 on 2025-01-06)

Response to the Referee comments on “ Simulation of the 1d XY model on a quantum computer ”
We are thankful for the referee’s detailed report and we consider it has helped to improve the readability of the new manuscript version as well as clarify some of its parts.
In the following lines, we reproduce the referee’s comments and respond to them providing, when appropriate, the explicit modifications on the new manuscript version.

Answer to report 2:
Comments on the text:
“I would suggest a revision of the text, as it is not very well-written in some parts. Here, I highlight some issues I found in the text, along with improved versions. However, I think there are many other issues throughout the text...
1. "In this paper, we present the comprehensive scheme for..." -> "In this paper, we present a comprehensive scheme for..."
2. "These transitions occur at absolute zero..." -> "These transitions occur at zero temperature..."
3. "The quadratic Hamiltonian in fermionic annihilation and creation operators appears in more condensed matter systems notably exemplified in the Hubbard model" -> "Hamiltonians quadratic in fermionic annihilation and creation operators are ubiquitous in condensed matter systems, describing systems of free fermionic particles." Concerning the Hubbard model: usually this model is considered with the interaction term, so is not quadratic...
4. "...leading us to the subsequent phase: the fermionic Fourier transform (fFT)." -> "Diagonalizing this type of Hamiltonian is a well-established process, achieved through the fermionic Fourier transform (fFT)."
5. "...our primary objective centers around obtaining the matrix" -> "...our primary objective is to obtain the matrix"
6. "Figure 3: In the diagram is shown the decomposition of the building block of Fnk shown in Eq.(60), where ϕk =−i2πkn." -> "Figure 3: The diagram illustrates the decomposition of the building block of Fnk (Eq.(60)), where ϕk =−i2πkn." “
We thank the referee for carefully proofreading our manuscript and helping us improve its readability. Accordingly, we have incorporated all the referee’s suggested changes and made several minor replacements to enhance clarity and readability while preserving the original intent of the text.

“Shouldn't the order of the three unitaries in Eq. (54) be reversed? Based on Eq. (52), it seems that U_JW should be applied to H first...”
As the referee rightly pointed out, we had mixed the correct order of operators in Eq.(52). The operator $U_{dis}=U_{BG} U_{FT} U_{JW}$ maps vectors from the spin basis to the eigenbasis of the Hamiltonian. Consequently, the diagonalized Hamiltonian, ($tilde{H}_{XY}$) is given by $ U_{dis} H_{XY} U^{dagger}_{dis} $ . This equation has now been corrected.

Other questions:
“Why you say the circuit is "specifically designed for the NISQ era"? What is NISQ specific here?”
We realized the term “specifically designed for the NISQ era” can be confusing and not clear and we have changed it to “suitable for the NISQ era”. While the circuit can also be used for Fault-Tolerant Quantum Computers (FTQC), its structure, which contains relatively few gates, makes it particularly advantageous for NISQ devices by helping to reduce the effects of noise in real hardware.
Moreover, the circuit serves as a valuable benchmark for NISQ systems. Additionally, its ability to implement time evolution allows for easy adjustment of complexity by adding more time evolution steps, enabling systematic hardware testing across different circuit depths, providing valuable insights into noise effects and error characterization, both of which are crucial for the NISQ era.

“Could your circuit serve as a starting point for a variational quantum circuit designed to address more complex, interacting Hamiltonians?”
As stated in our work, we believe that the use of this circuit does not necessarily have to remain limited to benchmarking purposes. The XY Hamiltonian, particularly in the 2D case, doesn’t present an analytical solution. Therefore, it could be interesting to use the circuit to simulate the 1D case as a base for constructing a more complex approach, such as finding the ground state of the 2D system.
One possible way to achieve this would be to introduce variational interactions within the circuit to approximate the effects of the 2D Hamiltonian that are absent in the 1D case. In fact, we have explored a similar approach, albeit using an adiabatic rather than a variational method. However, in our specific case, we encountered a gap crossing, which made unfeasible this approach.

Attachment:

referee_answer_SciPost_GZo4IQM.pdf

---

## Round 1 · Referee Report · Anonymous (Referee 1) · 2025-1-6

Report

The one-dimensional anisotropic XY model in transverse magnetic field is well known. Yet, even though some of materials rehash known facts, especially the part on Jordan Wigner transformation on spin and fermions, the article still reads well: The authors have so pedagogically presented the material that it makes reading enjoyable. I like it.

Section 3 presents detailed calculations and implementation of the simulation on a quantum simulator. The algorithm for fermionic Fourier transforms is based on work by Andrew Ferris (Ref. [21] in the article). And section 4 presents time evolution of the system. Section 5 presents results of simulations on a quantum simulator.

There may be missing references: previous work on such simulation exists in literature, e.g. Quantum Information Processing 20.8 (2021): 264; Physica Scripta 97.2 (2022): 025101; Physical Review Research 6.3 (2024): 033107, Physical Review A 95.5 (2017): 052339. The authors may wish to note some of these publications and cite them if needed.

Overall, I think the authors have made sufficient attempt to cast old wine in new wineskin. While I do not find anything novel in the article, I think this article still serves as a good pedagogical guide for beginners in quantum simulation. I would recommend acceptance after minor modifications.

Minor errors:

(i) Immediately after Eq(1), there should not be an indent after an equation. I think the authors left a space after the equation in their LaTeX version.
(ii) In the paragraph above Eq(2), "spin leather operators" should read "spin ladder operators".
(iii) Check Eq (3), there is an additional plus sign due to erroneous typing.

Recommendation

Ask for minor revision

  • validity: -
  • significance: -
  • originality: -
  • clarity: -
  • formatting: -
  • grammar: -

Author:  Marc Farreras  on 2025-03-11  [id 5275]

(in reply to Report 1 on 2025-01-06)

Response to the Referee comments on “ Simulation of the 1d XY model on a quantum computer ”
We are thankful for the referee’s detailed report and we consider it has helped to improve the readability of the new manuscript version as well as clarify some of its parts.
In the following lines, we reproduce the referee’s comments and respond to them providing, when appropriate, the explicit modifications on the new manuscript version.

Comments on the text:
“There may be missing references: previous work on such simulation exists in literature, e.g. Quantum Information Processing 20.8 (2021): 264; Physica Scripta 97.2 (2022): 025101; Physical Review Research 6.3 (2024): 033107, Physical Review A 95.5 (2017): 052339. The authors may wish to note some of these publications and cite them if needed.”
We sincerely appreciate the referee's suggestion regarding relevant references. After carefully reviewing the mentioned works, we find them highly relevant to our study. Consequently, we have expanded the introduction to emphasize the importance of using circuits with known exact solutions as essential benchmarks for mitigating experimental errors in real hardware. As these references highlight, analytical systems serve as ideal cases for this purpose. Accordingly, we have cited these works where appropriate to strengthen the context of our study.

“Minor errors: (i) Immediately after Eq(1), there should not be an indent after an equation. I think the authors left a space after the equation in their LaTeX version. (ii) In the paragraph above Eq(2), "spin leather operators" should read "spin ladder operators". (iii) Check Eq (3), there is an additional plus sign due to erroneous typing.”
We sincerely thank the referee for identifying these minor errors. We have corrected them as thoughtfully suggested.

Attachment:

referee_answer_SciPost.pdf

---

## Round 1 · Referee Report · Anonymous (Referee 3) · 2025-2-3

Report

The manuscript discusses a quantum circuit that embodies the exact diagonalization of the XY Hamiltonian in transverse field.

———

The topic is not novel per se, given that Ref. 7 has already laid out all the necessary ingredients some 15 years ago (and Ref. 21 has discussed fermionic Fourier transforms in more details a decade ago). Moreover, the last Author has already presented in Ref. 8 results obtained with the very same quantum circuit on IBM & Rigetti machines.

It would be desirable if the text would communicate in a much clearer way what is the element of novelty that the present study brings along -- one guess that I can make is that the quality of the experimental data seems to be considerably better and in excellent agreement with theory, but this should come out of the text without requiring the reader to skim through other papers.

I am glad to see if other points can be brought by the Authors.

———

The manuscript clearly aims at being very pedagogical, with lot of details about textbook calculations. Although I very much like the spirit in general, it seems to me that it wanders off into being excessively pedantic in many parts (see pages full of small intermediate manipulations of equations), while being imprecise in others (e.g., quoting the Hubbard model as an example of quadratic Hamiltonian… or stating that Eq.(52) converts the H_{XY} Hamiltonian in a non-interacting one…). It almost reads as the first version of a master thesis, that still needs a round of polishing.

By the way, I might have got lost with the notation, but I miss why the ground state of the XY model should not be in the fermionic half-filled sector (i.e., in the zero magnetization one when dealing with spins), but rather be given by Eq. (68). I am almost sure it is simply some glitch, but it proves that overwhelming the reader might also be counterproductive.

Same applies with the fact that the important discussion about periodic/antiperiodic boundary conditions depending on the fermionic population (the spin magnetization) is put aside at some point and only the even case seems to matter from there on…
Shrinking excessive details could offer a possibility of letting messages pass better.

In particular, some effort should be spent instead around Fig. 18 in explaining how a continuous (Ising) phase transition could ever exhibit discontinuities in the (transverse) magnetization, even more in a system of finite size… This looks more like some level crossing / first order phase transition, which does not sound right, but I possibly misinterpreted the figure caption.

———

As a stark contrast, the text feels really to come short when it deals with the actual results on the quantum computer and their discussion. The agreement between theory curves and experimental data looks excellent, which is a remarkable news in the realm of NISQ digital simulations at present date (compare the above mentioned Ref. 8 by the last Author). Therefore I would have naively thought that this was the central message of this work, while the section reads a bit sketchy and not delving into the reasons that allowed for such a nice performance. Along the same spirit, it would be nice to read some deeper consideration on the perspective of applying the same machinery to other models, possibly hinting at foreseen roadblocks to be circumvented and not just listing a couple of model names.

———

Overall I recommend the Authors to perform a major revision of their Manuscript to bring it up to the (fairly high) standards of the SciPost family, and to consider resubmitting it to the Lecture Notes series, if the element of novelty is less prominent than the pedagogical aim.

Recommendation

Ask for major revision

  • validity: good
  • significance: ok
  • originality: ok
  • clarity: good
  • formatting: reasonable
  • grammar: good

Author:  Marc Farreras  on 2025-03-11  [id 5277]

(in reply to Report 3 on 2025-02-03)

Response to the Referee comments on “ Simulation of the 1d XY model on a quantum computer ”
We are thankful for the referee’s detailed report and we consider it has helped to improve the readability of the new manuscript version as well as clarify some of its parts.
In the following lines, we reproduce the referee’s comments and respond to them providing, when appropriate, the explicit modifications on the new manuscript version.

Answer to report 3:
Comments on the text:
“The manuscript clearly aims at being very pedagogical, with lot of details about textbook calculations. Although I very much like the spirit in general, it seems to me that it wanders off into being excessively pedantic in many parts (see pages full of small intermediate manipulations of equations), while being imprecise in others (e.g., quoting the Hubbard model as an example of quadratic Hamiltonian… or stating that Eq.(52) converts the H_{XY} Hamiltonian in a non-interacting one…). It almost reads as the first version of a master thesis, that still needs a round of polishing.”
We acknowledge that using the Hubbard model as an example of a quadratic Hamiltonian could be confusing. We were specifically referring to the Fermi-Hubbard model in the weak interaction regime (with the interaction term set to zero). To avoid any misunderstandings, we have decided to remove that reference.
Regarding Eq. (52), we had incorrectly ordered the operators. The correct mapping from the spin basis to the eigenbasis of the Hamiltonian is given by the operator $U_{dis} = U_{BG} U_{FT} U_{JW}$, which ensures that the diagonalized Hamiltonian, $\tilde{H}{XY}$, is expressed as $\tilde{H}{XY} = U_{dis} H_{XY} U^{\dagger}_{dis}$. This equation has now been corrected, and we have revised our explanation to be clearer and more concise.
Additionally, we have made an overall effort to improve the readability of the text, reducing excessive textbook calculations and ensuring that the most potentially confusing points are explained more clearly to the reader.

“By the way, I might have got lost with the notation, but I miss why the ground state of the XY model should not be in the fermionic half-filled sector (i.e., in the zero magnetization one when dealing with spins), but rather be given by Eq. (68). I am almost sure it is simply some glitch, but it proves that overwhelming the reader might also be counterproductive.”
In Eq. (68), we analyze the ground state of the symmetric XY model, where the anisotropy parameter is $\gamma = 0$. This ensures symmetry between the X and Y axes. In this particular case, the Hamiltonian becomes diagonal immediately after applying the Fourier transform. From our expression, the eigenstate with the lowest eigenvalue corresponds to a state with a single fermion in the $k=2$ mode.
This result differs from the expected result of the symmetric XY model, where the ground state belongs to the half-filled sector because we impose periodic boundary conditions in the fermionic space rather than in the spin space, which is the more typical case. In principle, boundary conditions do not affect the system in the thermodynamic limit. However, for finite-size systems, they can significantly alter the energy values of each state.

“Same applies with the fact that the important discussion about periodic/antiperiodic boundary conditions depending on the fermionic population (the spin magnetization) is put aside at some point and only the even case seems to matter from there on…
Shrinking excessive details could offer a possibility of letting messages pass better.”
We appreciate the feedback regarding the confusion about boundary conditions. We have reviewed the text and revised several sections to clearly emphasize that this circuit is designed for $2^n$ qubits, where $n$ is an integer, as this is the case in which the fermionic Fourier transform can be performed efficiently.
Additionally, we are not using the standard XY Hamiltonian but rather imposing boundary conditions that correspond to periodic boundary conditions (PBC) in the spinless fermionic modes. In contrast to the standard case, where one must analyze how PBC in the spin operators translate into either PBC or antiperiodic boundary conditions (APBC) depending on the spin magnetization, here we can work with PBC independently of the spin magnetization.
Furthermore, since we consider only an even number of qubits (or equivalently, an even number of sites), the Fourier transform does not require analyzing the values of $k$ for an odd number of sites. Beyond this point, the boundary conditions no longer need to be carefully handled.

“In particular, some effort should be spent instead around Fig. 18 in explaining how a continuous (Ising) phase transition could ever exhibit discontinuities in the (transverse) magnetization, even more in a system of finite size… This looks more like some level crossing / first order phase transition, which does not sound right, but I possibly misinterpreted the figure caption.”
The boundary conditions used in this work differ from the conventional periodic boundary conditions (PBC) typically imposed in spin Hamiltonians. Instead, we adopt boundary conditions that circumvent the common issue arising from the Jordan-Wigner transformation, where the periodic/antiperiodic nature of the fermionic operators depends on the fermionic population. Specifically, our boundary conditions adapt based on whether the spin population contains an even or odd number of spin-up states. After applying the Jordan-Wigner transformation, these conditions always map to PBC for the fermionic operators, not requiring us to account for different fermionic populations. Consequently, our Hamiltonian approximates the expected behavior in the thermodynamic limit, i.e. when lambda=1 it exhibits a discontinuity in the magnetization. However, as the number of particles increases, the influence of the boundary conditions decreases, recovering the expected behavior of the original XY model Hamiltonian. This effect is evident in the following figure (see the attach PDF file), where we have performed an exact diagonalization of the Hamiltonian used in our work to obtain the ground state and compute its magnetization.

As a stark contrast, the text feels really to come short when it deals with the actual results on the quantum computer and their discussion. The agreement between theory curves and experimental data looks excellent, which is a remarkable news in the realm of NISQ digital simulations at present date (compare the above mentioned Ref. 8 by the last Author). Therefore I would have naively thought that this was the central message of this work, while the section reads a bit sketchy and not delving into the reasons that allowed for such a nice performance. Along the same spirit, it would be nice to read some deeper consideration on the perspective of applying the same machinery to other models, possibly hinting at foreseen roadblocks to be circumvented and not just listing a couple of model names.

To be clear, there are no experimental results shown in this article. The points shown in Fig. 18 and 19 correspond to a simulation of the quantum circuit, to show the agreement with the theory. The error bars correspond to the statistical error of that simulation (in other words, we simulate the quantum circuit assuming perfect quantum gate operations and a specific number of measurements). We have clarified this point in the sentence "arios. The results show the classical simulation using the quantum computing library Qibo [REF], for the spin chain n = 4 and n = 8 using the circuits represented in Figs. 13, 14. “
With current coherence times and gate errors of IBM's quantum computers, we expect similar results as the ideal simulation, since this is an n=4 qubit shallow circuit. The use of these quantum computers has increased almost exponentially in the last years, which makes really difficult for free users to use the devices for scientific projects without paying a subscription (very long execution queues). Since this circuit was already executed in 2018, we decided not to push for a revision of the experimental results (otherwise it will have take much longer to finalize it), and focus our work in a pedagogical revision of the simulation of XY model with the corresponding code that can be used for the community on IBM's and other quantum computers.
We also extended the conclusions to discuss a little bit more the perspective of extending this work to study other condensed matter models.
Other questions:
It would be desirable if the text would communicate in a much clearer way what is the element of novelty that the present study brings along -- one guess that I can make is that the quality of the experimental data seems to be considerably better and in excellent agreement with theory, but this should come out of the text without requiring the reader to skim through other papers. I am glad to see if other points can be brought by the Authors.
The key contributions of our work include an open-source code that enables the construction of a disentangling circuit for the 1D XY model, generalized for any number of qubits of the form $2^n$, where $n$ is an integer. Additionally, our implementation is based on the Qibo framework, whereas previous studies on simulating the 1D XY model primarily relied on Qiskit.
Furthermore, we introduce a circuit for performing exact time evolution, a feature not proposed in prior works. However, we acknowledge that the primary objective of this work is pedagogical. In this regard, we agree that it would be more appropriately suited for SciPost Lecture Notes or Physics Core rather than SciPost Physics.

Attachment:

referee_answer_SciPost_Cx1w9aj.pdf

---

## Editorial Decision

resubmitted